



# Debris cover and the thinning of Kennicott Glacier, Alaska, Part A: in situ mass balance measurements

Leif S. Anderson[1,2], Robert S. Anderson[1], Pascal Buri[3], and William H. Armstrong[1,4]

[1]Department of Geological Sciences and Institute of Arctic and Alpine Research, University of Colorado Campus Box 450, Boulder, CO 80309, USA
[2]GFZ German Research Centre for Geosciences, Telegrafenberg, 14473 Potsdam, Germany
[3]Geophysical Institute, University of Alaska-Fairbanks, 2156 Koyukuk Drive, Fairbanks, AK 99775, USA
[4]Department of Geological and Environmental Sciences, Appalachian State University,033 Rankin Science West, ASU Box 32067, Boone, NC 28608-2067, USA

*Correspondence to*: Leif Anderson (leif@gfz-potsdam.de)

**Abstract.** The mass balance of many Alaskan glaciers is perturbed by debris cover. Yet the effect of debris on glacier response to climate change in Alaska has largely been overlooked. In three companion papers we assess the role of debris, ice dynamics, and surface processes in thinning Kennicott Glacier. In Part A, we report in situ measurements from the glacier surface. In Part B, we develop a method to delineate ice cliffs using high-resolution imagery and produce distributed mass balance estimates. In Part C we explore feedbacks that contribute to glacier thinning.

Here in Part A, we describe data collected in the summer of 2011. We measured debris thickness (109 locations), sub-debris melt (74), and ice cliff backwasting (60) data from the debris-covered tongue. We also measured air-temperature (3 locations) and internal-debris temperature (10). The mean debris thermal conductivity was 1.06 W (m C)$^{-1}$, increasing non-linearly with debris thickness. Mean debris thicknesses increase toward the terminus and margin where surface velocities are low. Despite the relatively high air temperatures above thick debris, the melt-insulating effect of debris dominates. Sub-debris melt rates ranged from 6.5 cm d$^{-1}$ where debris is thin to 1.25 cm d$^{-1}$ where debris is thick near the terminus. Ice cliff backwasting rates varied from 3 to 14 cm d$^{-1}$ with a mean of 7.1 cm d$^{-1}$ and tended to increase as elevation declined and debris thickness increased. Ice cliff backwasting rates are similar to those measured on debris-covered glaciers in High Mountain Asia and the Alps.


**Keywords** ice cliff; backwasting; lapse rate; meteorology; melt; thermal conductivity

## 1 Introduction

Debris cover is common on valley glacier surfaces (Scherler et al., 2018) and when thick it suppresses melt rates. As climate warms, glaciers with and without debris thin. But counterintuitively many debris-covered glaciers thin most rapidly where melt rates should be suppressed by thick debris. This phenomenon is known as the 'debris-cover anomaly' (Pellicciotti et al., 2015), which could be caused by *melt hotspots* like ice cliffs or surface lakes within the debris cover (e.g., Sakai et al., 2000; Miles et al., 2018). Or alternatively by reduced ice flow from upglacier which we refer to as *dynamic thinning* (Vincent et al., 2016; Brun et al., 2018).



In three companion papers we constrain patterns of ice cliff and sub-debris melt, ice dynamics, and surface processes to explain the debris-cover anomaly occurring on Kennicott Glacier, Alaska. In addition to helping understand the causes of the debris-cover anomaly, Kennicott Glacier has several aspects that warrant study.

First, Kennicott Glacier exists at a latitude (61.5° N), climate (sub-polar) and in a region (NW North America) where the effects of debris cover has largely been neglected, aside from a few pioneering studies (Loomis, 1970; Driscoll, 1980;
Mattson, 2000). Glaciers are very sensitive to the pattern of melt under debris (Anderson and Anderson, 2016), but actually measuring melt across debris-covered glaciers is difficult. It requires that the effects of debris and *melt hotspots* be constrained, which often requires abundant in situ measurements and instrumentation.

Partly because of the significant effort required to make in situ measurements, mass balance research of debris-covered glaciers has been focused on a few keystone glaciers in the Himalaya (e.g., Lirung, Ngozumpa, and Khumbu Glaciers; Benn
et al., 2012; Immerzeel et al., 2014) and European Alps (e.g., Miage and Zmutt Glaciers; Brock et al., 2010; Mölg et al., 2019). Sparse in situ observations mean that global projections of glacier change cannot yet robustly incorporate the effects of debris cover. Measurements from debris-covered glaciers in new regions like Alaska are therefore needed. In order for debris-covered mass balance models to be applied regionally, basic debris properties and the meteorology above the debris must also be measured.

Second, Kennicott Glacier is covered by thinner debris than most previously studied glaciers. Thinner debris means that sub-debris melt rates will be higher further increasing the likelihood that anomalous glacier thinning can be explained by *melt hotspots* instead of *dynamic thinning*.

Third, Kennicott Glacier hosts a higher concentration of ice cliffs within its debris cover than any previously studied glacier (see Part B). Ice cliff backwasting counters the insulating effect of debris (e.g., Sakai et al., 2002). The combination of
relatively thin debris and an abundance of ice cliffs increases the likelihood that *melt hotspots* will compensate for the insulating effects of debris to produce average melt rates simular to debris-free glaciers. To date no theory exists for how ice cliff distribution or backwasting rate varies in space or time. Kennicott Glacier offers a chance to explore a large population of ice cliffs over a range of elevations and debris thicknesses, which may help us understand the controls of ice cliff distribution.

Here, in Part A we describe an array of in situ measurements. We present debris thickness, as well as sub-debris and ice cliff melt data. We also measure debris thermal properties and air temperature above the glacier surface. These data lay a foundation upon which future research on Kennicott Glacier can build: they are prerequisite for our distributed melt estimates and our presentation of feedbacks that contribute to glacier thinning, presented in Parts B and C.

### 1.1 Study Glacier

Kennicott Glacier is located on the south side of the Wrangell Mountains between 4996 m and 400 m elevation (Fig. 1; 4600 m elevation range; 387 km² area). For comparison, Khumbu Glacier, in Nepal, has an area of 26.5 km² and spans an elevation range of 3950 m from 8848 m to about 4900 m a.s.l. Kennicott Glacier covers almost 15 times more area than the



Khumbu Glacier and our study area, the debris-covered tongue of Kennicott Glacier, is 24.2 km$^2$, only slightly smaller than Khumbu Glacier.

In total 20% of Kennicott Glacier is debris-covered. At elevations below the equilibrium-line altitude at about 1500 m (Armstrong et al., 2017), 11 medial moraines are identifiable. These medial moraines form primarily from the erosion of hillslopes above the glacier and express themselves as stripes on the glacier surface (see Anderson, 2000). Above 700 meters elevation, debris is typically less than 5 cm thick or roughly one clast thick. The medial moraines coalesce in the last 7 km of the glacier where ice cliffs, surface lakes, and streams are scattered within otherwise continuous debris cover.

Mount Blackburn, the highest peak in the Wrangell Mountains, is situated in the Kennicott Glacier watershed between the Yukon river basin to the north and Copper River basin to the south. The main trunk of Kennicott Glacier is 42 km long and is joined by two primary tributaries, the Root and the Gates Glaciers. Kennicott Glacier has only retreated 600 meters since its maximum Little Ice Age extent in 1860, although it has thinned considerably, even under thick debris (Parts A and B; Rickman and Rosenkrans, 1997; Das et al., 2014; Larsen et al., 2015).

## 2 Data collection and Results

We conducted a two-month field campaign between June 18 and August 16 2011. Because Kennicott Glacier debris has not been previously studied, we made a wide range of in situ measurements. We measured air temperature above the glacier surface, debris thickness, sub-debris melt rates, internal debris temperature, debris surface temperature, and ice cliff backwasting rates (Figs. 2 and 3). Our measurements were made in the terminal region of the glacier where the medial
moraines join to form a continuous debris cover across the glacier. Our sampling sites range between 700 and 450 meters elevation.

### 2.1 Air temperature above debris cover

Boundary layer conditions vary widely between debris-free and debris-covered glaciers (e.g., Brock et al., 2010). This is primarily due to differences in surface temperature between debris-free and debris-covered glacier surfaces. During the melt
season, debris-free surface temperatures are typically near the melting point. But debris surface temperatures vary strongly with debris thickness and surface topography (e.g., Shaw et al., 2016).

We installed three air temperature poles (Fig. 2; at 513, 600, and 704 m a.s.l.) to document the interplay between debris, air temperature, and surface melt across the study area. Air temperatures were measured hourly between July 21 and August 9, using iButton thermistors in standard HOBO radiation shields. Poles were partially drilled into the ice and then debris 30 cm
thick was accumulated around the pole in a 1 meter radius. Thermistors were initially placed 1.8 m above the debris surface. Thermistors were moved back to their original elevation above the debris surface in the middle of station deployment. The maximum deviation of thermistors from the original height above the surface was 14 cm. The thermistors have a measurement uncertainty of 0.5 °C based on ice bath calibration. We refer to the individual temperature poles as the Lower, Middle, and Upper weather stations (*LWS*, *MWS*, *UWS*). Unperturbed debris thicknesses averaged 2 cm at the *UWS*, 10 cm
at the *MWS*, and 22 cm at the *LWS*. The *UWS* and *MWS* were placed in areas with lower surface relief while the *LWS* was located at the base of a topographic bulge at an important transition on the glacier surface (see Part C).





Figure 4 shows data from the *LWS* as well as near-surface air temperature lapse rates (*LR*). *LR*s between *LWS* and *MWS* were extremely steep, up to -74 C km$^{-1}$, with a mean *LR* of -25 C km$^{-1}$ over the measurement period (Table 1). *LR*s were more shallow between the *MWS* and *UWS*, with a mean of -7.3 C km$^{-1}$, more typical of *LR*s above debris cover (-7.5 C km$^{-1}$,

Mihalcea et al., 2006; -8.0 and -6.7 C km$^{-1}$, Brock et al., 2010; -5 to -7.8 C km$^{-1}$, Steiner and Pellicciotti, 2016).

The study area is spanned by off-ice meteorological stations, which are typically used for estimating melt on glacier surfaces. Gates Glacier meteorological station is located at 1240 m a.s.l. and May Creek meteorological station is located 15 km to the southwest of McCarthy at 490 m a.s.l. (Fig. 1). For comparison, *LR*s calculated between these off-ice meteorological stations was a more typical -5.6 C km$^{-1}$ (e.g., Minder et al., 2010).

*LR*s tended to be steeper during the afternoon and evening than the night and morning (Table 1 and Fig. 4). An observation similar to those of Fujita and Sakai (2000) and Steiner and Pellicciotti (2016) on Lirung Glacier, Nepal and Brock et al. (2010) and Shaw et al. (2016) on Miage Glacier, Italy. The steepest *LR*s occurred between 5 and 7 pm. Steep *LR*s in the afternoon and evening are likely caused by higher debris surface temperatures, and convective heating at lower elevations. The daily-*LR* cycle was amplified on clear-sky days (Fig. 4; e.g., day-of-the-year (DOY) 203 and 204 compared to DOY

120 212-215).

Variable, steep *LR*s occurred on clear-sky days, while *LR*s were less variable and shallower on cooler, cloudy days. Amplified debris surface temperatures, as well as a decreased sky-view (see Steiner and Pellicciotti, 2016) at *LWS* may explain the extreme temperature differences between *LWS* and *MWS*. *UWS* and *MWS* are more likely to be influenced by cold-air drainage from up valley. Because debris is thicker at *LWS* debris surface temperatures will also tend to be higher.

These extreme air temperature differences highlight the need for further study of the relationship between debris, surface temperature, and air temperature variations on debris-covered glaciers.

## 2.2 Debris thickness

We documented debris thicknesses at 109 locations at the same locations we also measured ice cliff backwasting, sub-debris melt, and debris surface temperature (Fig. 2). We measured thicknesses by digging through the debris to the ice surface

(after Zhang et al., 2011). Where debris was thinner than ~10 cm we dug 5 pits and recorded the average debris thickness. While we did not measure debris thickness below 450 m elevation, visual inspection from across the proglacial lake suggests that debris exceeded 1 m above some ice cliffs.

Figure 5 shows debris thickness as it varies with elevation. Debris thickness tends to increase downglacier and varies from less than a few millimeters above 700 m elevation to as high as 1 meter above an ice cliff at 475 m a.s.l. (Table 2).

Transversely across the glacier, mean debris thicknesses tended to be larger near the glacier margin (Fig. 6). Debris greater than 40 cm thick was present near the eastern glacier margin between 650 and 700 m elevation. Debris up to 1 m thick was observed near the western margin at 700 m elevation. Toward the glacier interior and between 650 and 700 m elevation debris thickness did not exceed 15 cm.



### 2.3 Sub-debris melt

We measured sub-debris melt at 74 locations (Fig. 2). At each site, we removed debris, installed ablation stakes and then replaced the debris (Fig. 3). We placed stakes in debris up to 40 cm thick. Sub-debris melt ( $\dot{b}_{debris}$ ) was measured by removing the debris and measuring ice surface lowering.

Because melt measurements were made over different time periods we corrected each measurement to represent the full measurement period. A melt factor for sub-debris melt $MF_{debris}$ was therefore calculated for each measurement:


$$MF_{debris} = \frac{\sum_{i=1}^{n} \dot{b}_{debris}}{\sum_{i=1}^{n} T^{+} \Delta t} \quad (1)$$

where $T^{+}$ is the positive degree-days defined as the mean daily air temperature when above 0° C and $\Delta t$ is one day (e.g., Hock, 2003). Air temperatures did not drop below freezing during the study period. We use hourly air temperature data from the Gates Glacier and May Creek meteorological stations to estimate the $T^{+}$ at each site. The average timespan

between measurements was 24 days, the minimum was 8 days, and the maximum was 56 days. We refer to re-calculated sub-debris melt rates as expected melt rates.

Figure 7 shows the relationship between expected sub-debris melt rate and debris thickness (or Østrem's curve) during the study period. Highly variable melt rates beneath debris less than 3 cm thick prevented the establishment of a single relationship accounting for the melt-enhancing effects of thin debris (e.g., Østrem, 1959). For debris thicknesses less than 4

cm local conditions appear to be as important as debris thickness itself (see Mihalcea et al., 2006; Reid and Brock, 2010 for similar observations). Figure 8 shows that the relationship between melt rate and debris thickness from Kennicott Glacier is similar to those derived from other debris-covered glaciers at similar latitudes.

### 2.4 Debris temperature and thermal properties

In order to constrain debris thermal properties, thermistors were placed within the debris at ten sites. At each site, three to four iButton thermistors were placed in vertical profile within the debris (Fig. 2). All profiles were paired with nearby ablation stake measurements. Temperature was measured every 30 minutes and each profile was in debris for at least one week. Thermistors were placed in debris between 8 and 46 cm thick, in four medial moraines composed of distinct lithological combinations. The debris cover is composed of sedimentary and volcanic rocks, including: andesite,

pyroclastics, dacite, limestone, greenstone, and shale (MacKevett, 1972; MacKevett and Smith, 1972). We estimated debris conductivity using simultaneous measurements of sub-debris melt, internal debris temperature, and debris thickness (e.g., Nakawo and Young, 1981, 1982; Kayastha et al., 2000; Mihalcea et al., 2006). The heat flux ($Q_m$) is proportional to the effective thermal conductivity ($K_e$) of the debris for a given temperature gradient, which is set largely by surface temperature ($T_s$) due to the nearly constant 0° C summertime temperature of melting ice ($T_i$):






$$Q_m = K_e \frac{(T_s - T_i)}{h_{debris}} \quad , \, (2)$$

where $h_{debris}$ is debris thickness. The modifier *effective* is used to emphasize that heat is also transferred by advection of air and water (e.g., Juen et al., 2013). The sub-debris melt rate (per unit area) is related to the energy available for ablation ($Q_m$) through:

$$Q_m = L_f \rho_i a \quad , \, (3)$$

where $L_f$ is the latent heat of fusion of ice ($334 \times 10^3$ J kg$^{-1}$), $\rho_i$ is the density of ice (900 kg m$^{-3}$), and $a$ is the sub-debris melt rate (m d$^{-1}$). We estimated mean debris surface temperatures at each site by linearly extrapolating mean internal debris temperatures. Using Eq. (1 and 2), we then calculate $K_e$ for each temperature profile.

Debris thermal conductivity ranged from 0.525 to 2.16 W (m C)$^{-1}$ with a mean of 1.06 W (m C)$^{-1}$ (Fig. 9). Our estimates broadly agree with the range of previous direct measurements of $K_e$ from debris-covered glaciers (0.85 to 2.6 W (m C)$^{-1}$;

Nakawo and Young, 1982; Conway and Rasmussen, 2000; Juen et al., 2013). The measurements are also within the range of $K_e$ estimated from physical constants (0.47 to 1.97 W (m C)$^{-1}$; Nicholson and Benn, 2006). The apparent non-linear increase in debris conductivity with debris thickness, shown in Figure 9, maybe due to the presence of water (e.g., Mattson, 2000). Water increases bulk thermal conductivity and tends to increase thermal conductivity more in thick debris (see Nicholson and Benn, 2006 and data in their Table 2).

Following Conway and Rasmussen (2000), we also estimated debris thermal diffusivity (Fig. 10). We calculate the slope between the first derivative of internal debris temperature with depth and the second derivative of internal debris temperature with depth. Thermal diffusivity ranged from 0.067 to 0.76 mm$^2$ s$^{-1}$ with a mean of 0.32 mm$^2$ s$^{-1}$. These estimates are comparable to those made in other studies (0.6 and 0.9 mm$^2$ s$^{-1}$, Conway and Rasmussen, 2000; 0.3 and 0.38 mm$^2$ s$^{-1}$, Nicholson and Benn, 2006).

The thermal conductivity and diffusivity increase strongly with debris thickness, an observation supported by other studies (e.g., Kayastha et al., 2000; Nicholson and Benn, 2006; Mihalcea et al., 2006). Thermal diffusivity ($\kappa$) is directly proportional to thermal conductivity:

$$\kappa = \frac{K_e}{[C_{debris} \rho_{debris} (1 - \phi) + C_{void} \rho_{void} \phi]} \quad , \qquad (4)$$

where $\rho_{debris}$ and $C_{debris}$ are the density and specific heat capacity of the debris, $\varphi$ is the porosity, and $\rho_{void}$ and $C_{void}$ are the

density and specific heat capacity of the voids within the debris (see Nicholson and Benn, 2006). The increased scatter of thermal diffusivity compared to thermal conductivity is likely due to differences between the material properties of the debris from site-to-site and differences in uncertainty between the methods.

Debris thickness appears to be a primary control of debris conductivity, even though the thermistor profiles were placed in debris composed of distinct lithological mixes. Perhaps the increase in thermal properties with debris thickness reflects a

decrease in porosity as debris thickens. Thicker debris covers tend to have increased fine material at depth (e.g., Owen et al.,



2003), which tends to reduce porosity. If little air convection occurs in the pores of the thinner, higher porosity debris then the bulk $K_e$ would be greatly reduced due to the low thermal conductivity of air relative to rock (as described by Juen et al., 2013).

On a single clear-sky day, August 12, we measured debris surface temperature as it varied with debris thickness, across the upper portion of the study area (Fig. 2). We used a Fluke Infrared Thermometer to measure surface temperature. Thirty measurements from a representative area where made and averaged to produce each recorded data point. Figure 11 shows the surface temperature- debris thickness relationship. For debris thicknesses greater than about 20 cm, surface temperatures increased rapidly.

### 2.5 Ice cliff backwasting

Backwasting rates were measured at 60 ice cliffs. We made repeat horizontal distance measurements between the upper ice cliff edge and a stationary marker (in a moving reference frame; Fig. 3; after Han et al., 2010). Ice cliff backwasting rates were extrapolated to the full measurement period by calculating a melt factor for each ice cliff using data from the off-ice meteorological stations, as described for sub-debris melt above.

Figures 12 and 13 show that on average backwasting rates increased downglacier. Elevation-binned mean rates ranged from 215    5 cm d$^{-1}$ at 700 m to 9 cm d$^{-1}$ at 460 m. Within a single 50-meter elevation bin the standard deviation was a maximum of 4.7 cm d$^{-1}$ and a minimum of 2.0 cm d$^{-1}$. Ice cliffs backwasted at rates similar rates with and without ponds at their base.

On average, backwasting melt factors increased at lower elevations. Backwasting melt factors were slightly higher for ice cliffs facing northwest and slightly lower for those facing southeast (Fig. 13). Northwest-facing ice cliffs could have backwasted at higher rates due to the coincidence of warm afternoon air temperatures and afternoon and evening direct solar 220    radiation on clear-sky days. Southeast facing ice cliffs on the other hand receive their highest direct solar radiation fluxes in the morning when air temperatures are still cool, potentially explaining their lower melt factors. But these aspect differences in melt factor are small, contrasting with observations from glaciers at lower latitudes with more clear-sky days (e.g., Buri and Pellicciotti, 2018). This suggests that increased cloudiness and latitude lead to the lack of aspect control in ice cliff backwasting on Kennicott Glacier.

## 3 Discussion

### 3.1 Air temperature above debris cover

On sunny days, a steep air temperature lapse rate ($LR$) is observed above the debris. The extreme $LR$ s observed in the lower portion of the study area are likely caused by feedbacks related to debris thickness. On clear days, high direct solar radiation fluxes increase debris surface temperatures for debris thicker than ~20 cm much more than for thinner debris (Fig. 11). 230    Debris thicknesses at $LWS$ are in the range where debris surface temperatures rapidly increase with debris thickness. In contrast, at $MWS$ and $UWS$, debris thicknesses were only 10 and 2 cm, respectively. This suggests that strong gradients in air temperature may be related to feedbacks between debris thickness and surface temperature. The steep $LR$s could also be



related the large areas of exposed ice above the debris-covered tongue. The upper stations are more likely to be effected by descending katabatic winds. Air temperatures at the *LWS* may also be effected by the local debris-covered topography.

### 3.2 Debris thickness

Debris thickness varies between medial moraines (Fig. 6). Debris comprising the medial moraine furthest east on the glacier (immediately to the west of the town of Kennecott) shows considerably more thickness variability and larger mean thickness than the other moraines at the same elevation. This is consistent with the equations described in Anderson and Anderson (2018): thicker debris is more likely to occur at glacier margins where surface velocities are low compared to the

glacier interior. Surface lakes (see Part B) and debris cones are also common on this medial moraine which are rare at similar elevations in the glacier interior. Future distributed estimates of debris thickness on Kennicott Glacier or other glaciers with complex topologies and numerous medial moraines (e.g., the Baltoro Glacier, Pakistan) should consider extrapolation down individual flowlines.

Where ice is actively flowing, broad patterns of debris thickness can be changed dynamically by gradients in ice surface

velocity and debris melt out from under the debris. But in the lowest 4 kilometers of the glacier, where surface velocity gradients today are small (see Part C; Armstrong et al., 2016), debris thickness is likely primarily changing due to sub-debris emergence. But the melt-out of debris is limited by the fact that melt rates are reduced strongly as debris thickens (Fig. 7; Anderson and Anderson, 2018). The lowest 4 km of the glacier— that appears to not be actively deforming— was once underlaid by active ice (Rickman and Rosenkrans, 1997). Past velocity gradients would have helped define the current

debris thickness pattern even if those gradients are not apparent today. The debris thickness pattern in the lowest 4 km of the glacier is therefore likely a relict of past ice flow. Further debris thickness measurements and an analysis of the changes in ice dynamics through time are needed to further explore this hypothesis.

Debris thicknesses on glacier surfaces can vary by meters over 10-meter scales (e.g., Nicholson et al., 2018). Some of the scatter in our debris thickness measurements is derived from debris thickness variability caused by the local transport of

debris by mass wasting and other surface processes. The majority of our debris thickness measurements were derived from the top of ice cliffs. This potentially biases our measurements toward thinner values because surface debris tends to concentrate in topographic lows (e.g., Nicholson et al., 2018). Further debris thickness measurements should be made in topographic depressions, on the far western portion of the study area, and near the terminus. If our debris thickness measurements are biased toward thinner values then distributed estimates of surface melt using these debris thicknesses

would tend to over estimate sub-debris melt across the debris-covered tongue. This observation is important in Part B, where we compare distributed sub-debris melt estimates with the location of most rapid thinning on Kennicott Glacier, under thick debris.

### 3.3 Sub-debris melt

Measured sub-debris melt rates follow the typical shape of Østrem's curve from other glaciers (Fig. 8). But sub-debris melt

rates are higher for debris thicknesses greater than 15 cm, than on other debris-covered glaciers at similar latitudes. This discrepancy could be explained by the variability of air temperature within the study area: we measured sub-debris melt



across a 250 meter elevation range, and most melt measurements under thick debris were derived from the lowest portion of the study area. This difference could also reflect the proportion of fine material making up the thicker debris cover downglacier (e.g., Owen et al., 2003). Juen et al. (2013) demonstrated that increasing the proportion of fine material in debris covers increases water retention, ultimately increasing debris thermal conductivity and hence melt rates.

### 3.4 Ice cliff backwasting

In this study, backwasting was measured at the top of ice cliffs. Based on the modeling of Buri et al. (2016b) from Lirung Glacier, Nepal (28° N), the highest backwasting rates tend to occur near the top of ice cliffs. But making in situ measurements across a representative population of ice cliffs is very difficult. We assume that a single measurement from a number of ice cliffs would better represent the mean backwasting rate across the thousands of ice cliffs (Part B) on Kennicott Glacier. The validity of this assumption should be explored in future field campaigns. If it is true that ice cliff backwasting is maximized at the top of ice cliffs then distributed estimates of surface melt using our backwasting rates would tend to over estimate ice cliff backwasting. This observation is again important in Part B, where we estimate if ice cliff melt rates correspond to the location of maximum thinning under thick debris on Kennicott Glacier.

Ice cliff backwasting tends to increase toward lower elevations. This is linked to warmer air temperatures and increased ice cliff backwasting melt factors at low elevations. Potential causes for melt factor increase at lower elevations are: 1) a poor representation of air temperatures from off-glacier meteorological stations leading to an over estimation of the backwasting melt factor; 2) increased debris thickness proximal to ice cliffs at low elevations. Increased debris thickness leads to higher debris surface temperatures, longwave fluxes, and air temperatures (e.g., Brock et al., 2010); or 3) increased debris veneers and lower albedo of ice cliffs at low elevations (e.g., Reid and Brock, 2014). The portion of fine material making up debris covers tends to increase towards debris-covered glacier termini (Owen et al., 2003; Kellerer-Pirklbauer, 2008). These hypotheses require further data and analysis to test.

Thicker debris cover leads to higher debris surface temperatures, and higher longwave radiation fluxes received by ice cliffs. Despite this physical relationship, the backwasting rates measured on Kennicott Glacier are similar to those measured on glaciers with thicker debris cover and at lower latitude (Table 3). The similarity in backwasting rates suggests that there may be compensating effects between latitude, day length, and altitude. Ultimately the lack of aspect control on backwasting rates on Kennicott Glacier contrasts with observations from lower latitudes (e.g., Buri and Pellicciotti, 2018), suggesting that there may be a latitudinal control on ice cliff backwasting as it varies with orientation.

### 4 Conclusions

Strong air temperature gradients were documented above the debris cover of Kennicott Glacier, which are likely related to relationships between debris thickness, surface temperature and topography. Our observations highlight the need for further studies of micro-meteorology and its effect on the mass balance of debris-covered glaciers.

Debris thicknesses tend to increase downglacier. Despite rapidly increasing air temperatures downglacier, the insulating effect of debris dominates leading to reduced sub-debris melt toward the terminus. Transverse debris thickness patterns

broadly correspond with surface velocities such that in a single elevation band the thickest debris tends to occur near the glacier margin. The debris thickness data presented here can serve for the tuning and validation of distributed debris thickness estimates.

Internal debris temperature measurements constrain the thermal properties of debris. Conductivity and diffusivity increased

strongly with debris thickness. The non-linear relationship between conductivity and debris thickness suggests that water plays an important role in heat transfer through debris on Kennicott Glacier. These estimates of debris material properties, and their dependence on debris thickness, can be used as inputs to melt models.

Kennicott Glacier supports an unusually large population of ice cliffs (Part B), which counteract the insulating effects of thick debris. Our measurements of ice cliff backwasting varied from 3 to 14 cm $d^{-1}$ with a mean of 7.1 cm $d^{-1}$, similar to

backwasting rates from other debris-covered glaciers at starkly different elevations and latitudes. Backwasting rates tended to increase toward lower elevations, though significant scatter suggests that local conditions are important. The lack of aspect control on ice cliff backwasting rate strongly contrasts with observations from the Himalaya: there is likely a latitudinal control on the asymmetry of ice cliff backwasting and ice cliff survival. Data presented here are necessary for making distributed mass balance estimates on Kennicott Glacier and for further research of debris-covered glacier response

to climate change in Alaska.

**Data availability**

Datasets are available upon request.

**Author contribution**

LSA designed and funded the study, made field measurements, and composed the manuscript. RSA advised LSA and WHA during the process. PB provided integral comments and discussion points. WHA provided important comments and discussion points. All authors contributed to the writing of the manuscript.

**Competing Interests**

The authors declare that they have no conflict of interest.

**Acknowledgements**

LSA. acknowledges support from a 2011 Murie Science and Learning Center Fellowship, NSF DGE-1144083 (GRFP), and the COLD project awarded to Dirk Scherler. WHA acknowledges support from NSF OPP-1821002 the University of Colorado at Boulder's Earth Lab initiative. RSA. and WHA acknowledge support of NSF EAR-1239281 (Boulder Creek CZO) and NSF EAR-1123855. We thank Craig Anderson, Emily Longano, and Oren Leibson for field support. We thank

Regine Hock and Martin Truffer for thoughtful discussions. LSA thanks the organizers and participants of the 2010 Glaciological Summer School held in McCarthy, AK, which inspired this work. We thank Per Jenssen, Susan Fison, Ben Hudson, Patrick Tomco, Rommel Zulueta, the Wrangell-St. Elias Interpretive Rangers, the Wrangell Mountains Center,





Indrani Das, and Ted Scambos (NSIDC) for logistical support and the gracious loan of equipment. We thank Lucy Tyrell for facilitating outreach efforts.

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



**Tables**

**Table 1.** Near-surface air-temperature lapse rates in the study area

| Near-surface air temperature lapse rate ($LR$) [C km⁻¹] | Mean [C km⁻¹] | Stdev. [C km⁻¹] | Day [C km⁻¹] | Night [C km⁻¹] | Estimated debris surface temperature difference on clear-sky day [C] |
|---|---|---|---|---|---|
| Lower-Middle | -25.2 | 12.8 | -30.0 | -22.8 | 24 |
| Middle-Upper | -7.30 | 10.4 | -13.7 | -3.90 | 2 |
| Lower-Upper | -15.4 | 8.50 | -21.1 | -7.20 | 26 |
| May Creek – Gates Glacier | -5.59 | 3.98 | -8.95 | -3.85 | - |

*Lower, Middle, and Upper air temperature poles are on the glacier, May Creek and Gates Glacier stations are off-glacier.
**Estimated debris surface temperatures are based on data from Fig. 11.


**Table 2.** Statistics of debris- and melt-related variables.

| Measured variable | Mean | Std. | Minimum | Maximum |
|---|---|---|---|---|
| Debris thickness [cm] | 13.7 | 13.9 | 0.001 | 100 |
| Sub-debris ablation [cm d⁻¹] | 4.0 | 1.8 | 0.8 (37 cm of debris) | 7.3 (1 cm of debris) |
| Ice cliff backwasting [cm d⁻¹] | 7.1 | 2.5 | 2.8 | 13.8 |
| Ice cliff melt factor [cm (C d)⁻¹] | 10.9 | 3.6 | 4.6 | 21.1 |

**Table 3.** Comparison of ice cliff backwasting rates and debris thicknesses

| Glacier | Region | Latitude [deg.] | Mean study area elevation [m] | Range of backwasting rates [cm d⁻¹] | Mean debris thickness [cm] | Reference |
|---|---|---|---|---|---|---|
| Kennicott | Alaska, USA | 61 | 600 | 3-11 | 13 | This study |
| Miage | Alps, Italy | 46 | 2200 | 6.1-7.5 | 26 | Reid and Brock (2014) |
| Koxkar | Tien Shan, China | 42 | 3500 | 3-10 | 53 | (Han et al., 2010; Juen et al., 2014) |
| Lirung | Himalaya, Nepal | 28 | 4200 | 7-11 | 50-100 | (Buri and Pelicciotti, 2018) |
| Changri Nup | Himalaya, Nepal | 28 | 5400 | 2.2-4.5 | - | (Brun et al., 2018) |








# Figures

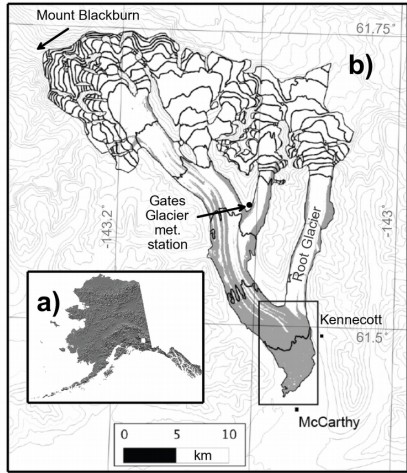

**Figure 1.** Map of Kennicott Glacier. a) Map of Alaska showing the location of panel b). b) Kennicott Glacier with Gates Glacier meteorological station (1240 m a.s.l.). Gates Glacier meteorological station is located at 1240 m elevation and the May Creek meteorological station is located at 490 m located 15 km to the southwest of the town of McCarthy.






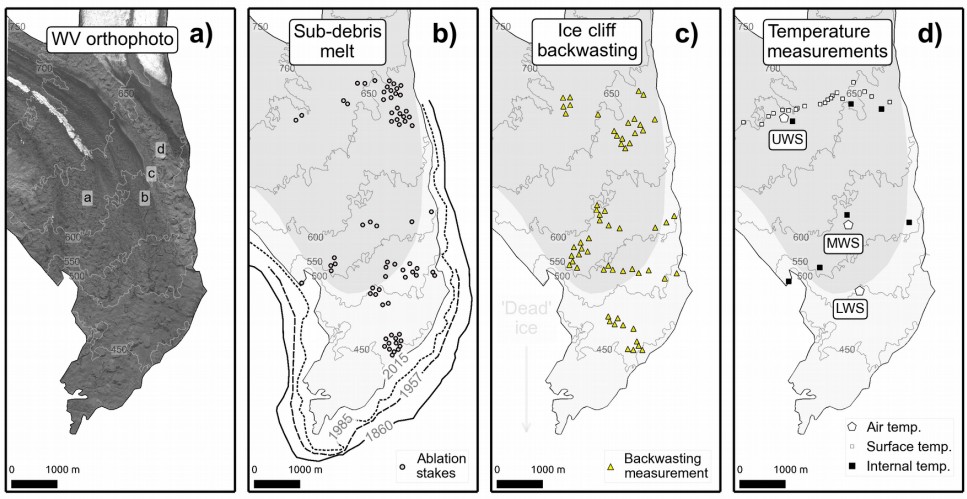

**Figure 2.** Satellite image of the study area with measurement locations. a) WorldView orthoimage from 2009 showing the study area. The letters in white boxes correspond to the medial moraines and panels in Fig. 6. b) Map of the study area ( 24.2 km²) with contours derived from ASTER GDEM V2 (2009). The ice extents are derived from WV imagery and aerial photos. The 'dead' ice portion of the debris-covered tongue is shown in light grey. It is defined by areas that only have measured daily mean surface velocities above 5 cm d⁻¹ during sliding events (Armstrong et al., 2016) and the observations of Rickman and Rosenkrans, 1997. c) Locations where ice cliff backwasting was measured. d) Location of air temperature poles, clear-day surface temperature measurements, and internal debris temperature profiles. The letters label the air temperature poles: Upper weather station (*UWS*); Middle weather station (*MWS*); Lower weather station (*LWS*).





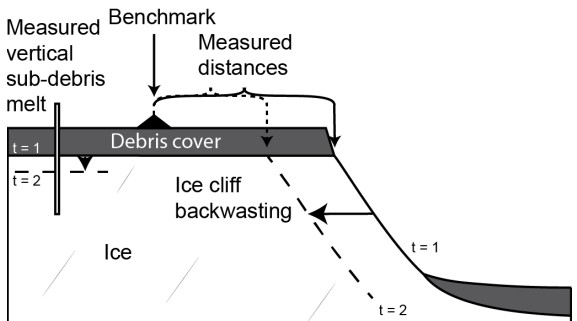

**Figure 3.** Schematic of field measurements.



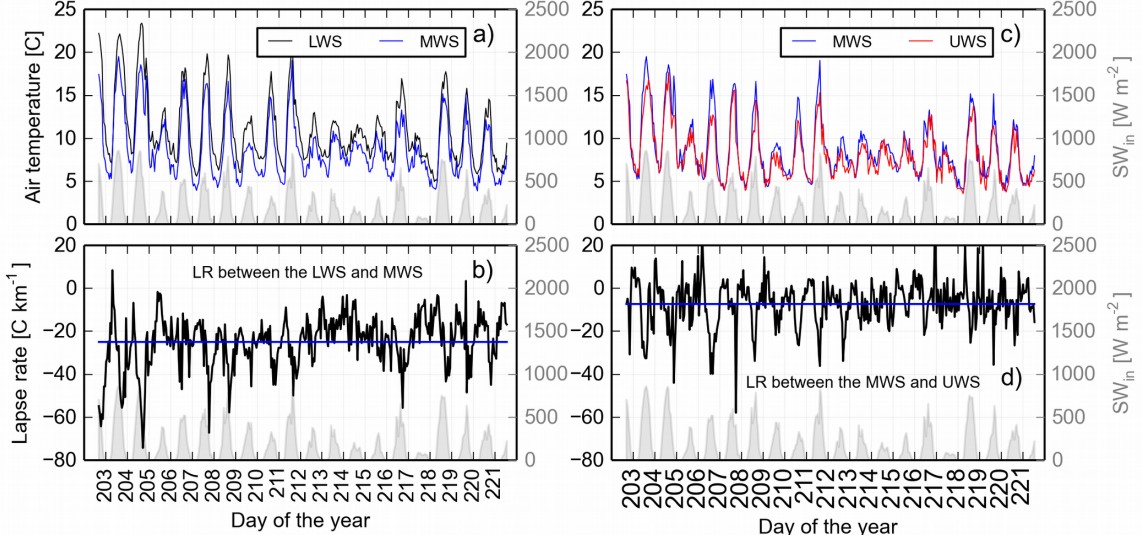

**Figure 4.** Air temperature and near-surface air temperatures lapse rate (*LR*) within the study area. a) Screen-level air temperature data from Lower weather station (*LWS*), Middle weather station (*MWS*), and incoming shortwave radiation data from the May Creek meteorological station. b) *LR* between *LWS* and *MWS* with May Creek shortwave radiation. c) Screen-level air temperature from *MWS*, Upper weather station (*UWS*), and incoming shortwave radiation data recorded from May Creek meteorological station. d) *LR* between *MWS* and *UWS* with May Creek shortwave radiation.






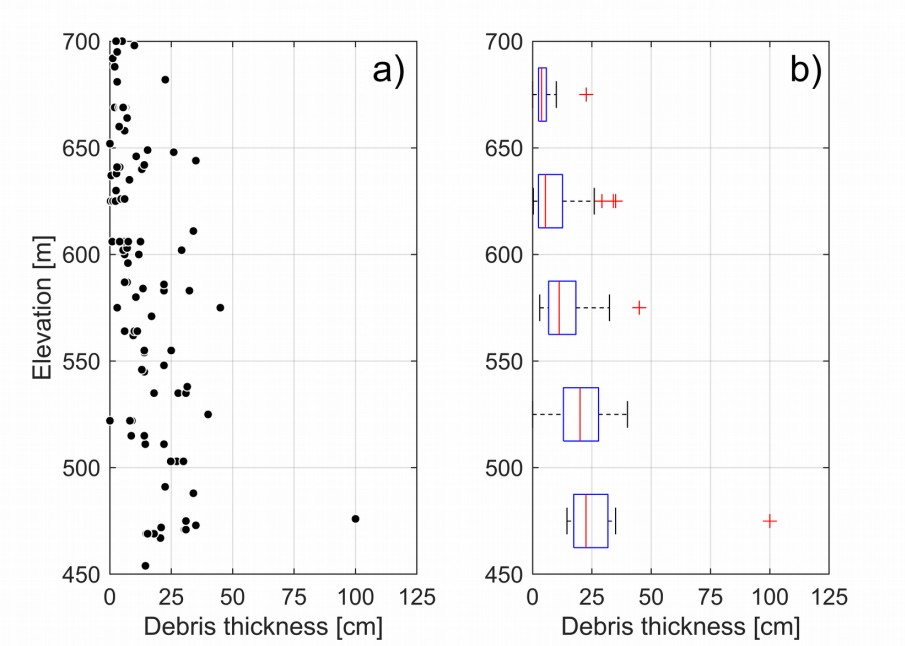

**Figure 5.** Pattern of debris thickness with elevation. a) In situ debris thickness measurements. b) Debris thickness boxplots in 50 meter elevation bins. Outliers are represented as +'s.






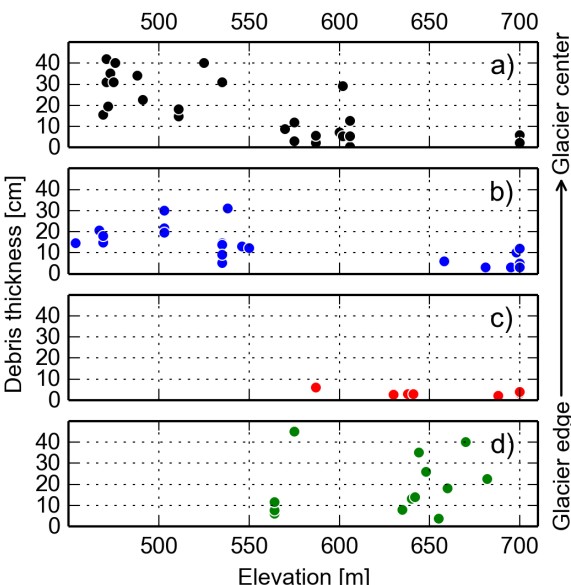

**Figure 6.** Debris thicknesses from different, coalesced medial moraines. See Figure 2 for the location of the corresponding medial moraines represented with each panel. Panel d) shows data from the medial moraine closest to the eastern glacier edge. The glacier interior is closer to panel a).





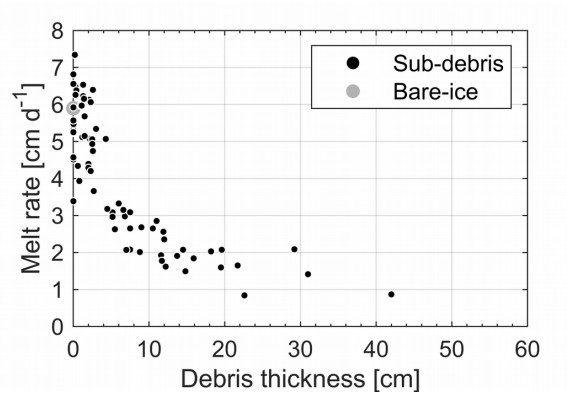

**Figure 7.** Dependence of melt on debris thickness also referred to as Østrem's curve. Sampling locations are shown in Figure 2.









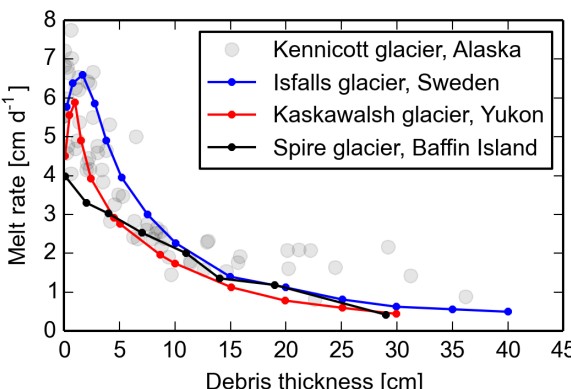

**Figure 8.** Comparison between Kennicott (61.5° N) sub-debris melt rates and other glaciers at similar latitudes. Isfallsglaciären (67.9° N; Østrem, 1959), Kaskawalsh Glacier (60.8° N; Loomis, 1970), Spire Glacier (66.4° N; Crump et al., 2017).









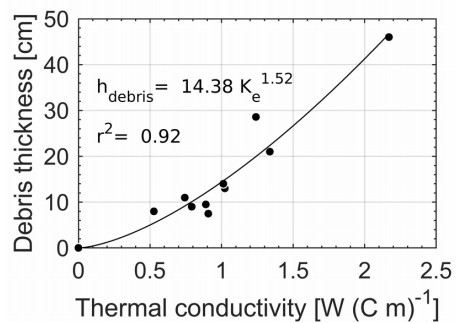

**Figure 9.** Thermal conductivity of the debris cover. a) Effective thermal conductivity ($K_e$) as it varies with debris thickness. Estimates are based on internal debris temperatures and sub-debris melt measurements over at least a week.





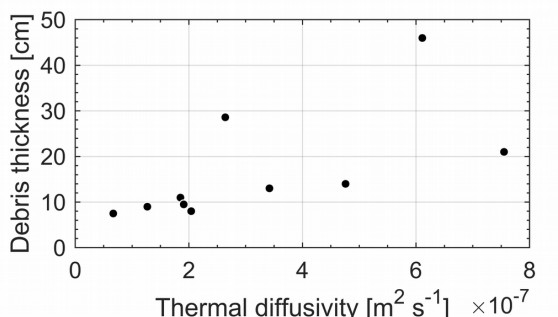

**Figure 10.** Dependence of thermal diffusivity on debris thickness. Estimates are based debris temperatures from 10 thermistor profiles using the method described by Conway and Rasmussen (2000).






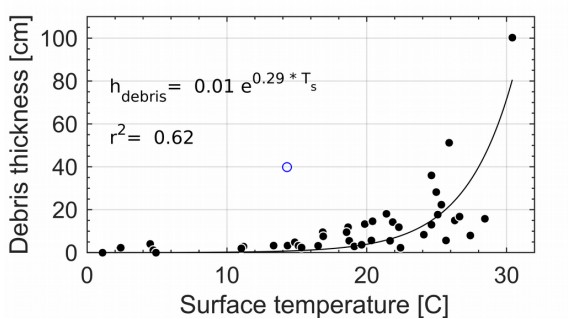

**Figure 11.** Maximum surface debris temperature from a single clear-sky day, August 12, 2011. The outlier shown in blue was shaded by local glacier-surface topography.





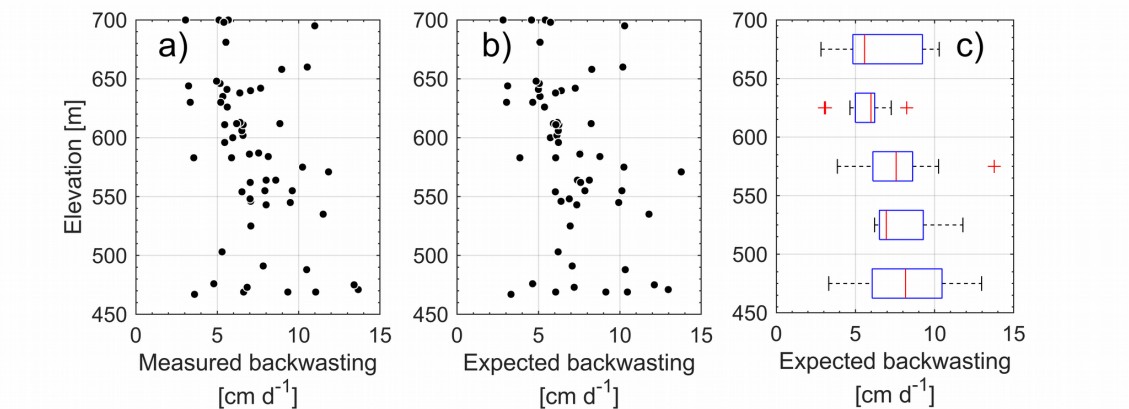

**Figure 12.** Pattern of ice cliff backwasting as it varies with elevation. a) Measured ice cliff backwasting over different time intervals. b) Expected ice cliff backwasting based the individual melt factor for each ice cliff. c) Expected backwasting boxplots in 50 meter elevation bins. Outliers are represented as +'s.







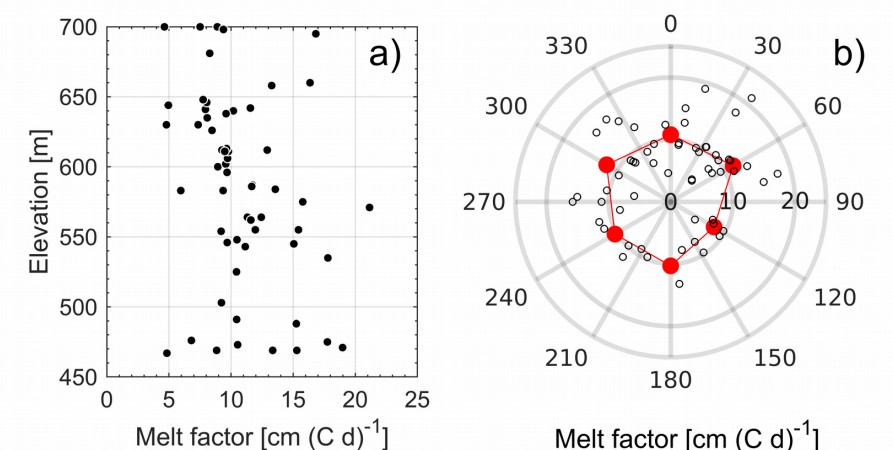

**Figure 13.** Ice cliff melt factors. a) Ice cliff melt factor as it varies with elevation. b) Ice cliff melt factor as it varies with aspect. The red dots represent the mean melt factor from 60° bins.