# Peer review of "Debris cover and the thinning of Kennicott Glacier, Alaska, Part A: in situ mass balance measurements"

_The Cryosphere, 2019_

## Referee Comment (RC1) · David Rounce (Referee) · 16 Oct 2019

**Review of "Debris cover and the thinning of Kennicott Glacier, Alaska, Part A: in situ mass balance measurements" by Anderson et al.**

This study is the first part of three publications that investigate debris cover on Kennicott Glacier in Alaska. Given the limited number of studies that measure properties and melt rates of debris-covered glaciers, these measurements and results are important for advancing our understanding of debris-covered glaciers. This is especially true when one considers the limited knowledge of debris-covered glaciers in Alaska. The measurements and results are presented well. For the most part, the study is easy to follow, well-written, and has sufficient references.

There are a few sentences/paragraphs that could be modified to improve their readability though. The only major comment is to make sure that this study is discussing results that specifically pertain to this part of the three-part study. There are also a couple places where additional detail or analysis would provide useful context to the modeling community; however, this would only require minimal additional work. Therefore, I recommend accepting this manuscript for publication subject to minor revisions. Please see my detailed comments below.

Main Comments

The reasons for studying Kennicott Glacier largely come across as reporting results across the three papers as opposed to stating what each paper does. For example, L55-57 state that the debris is thinner than most previously studied, but there is no reference to any studies concerning debris thicknesses on Kennicott Glacier. Similarly, L58 states there are more ice cliffs than those previously studied without a reference to a study that shows this. Hence, these appear to be results (and results from other papers) that are stated in the introduction.

Furthermore, the introduction states multiple times that the thinner debris increases the likelihood that melt hotspots will compensate for the insulating affects; however, thinner debris has higher melt rates, so it's unclear why melt hotspots would be more important for debris-covered glaciers with thinner debris because there would be less contrast between the sub-debris and ice cliff melt rates. If this is a hypothesis, then please state it this way.  If this is supported by a physical basis, then please explicitly state this reasoning.

Lastly, the interpretation of the transverse variations of debris thickness appear to be poorly supported by the present figures and text. L135-139 state that mean debris thicknesses increase near the glacier margins. However, site a appears to be closest to the center of the glacier, yet it has thicker debris. Similarly, site c is between sites b and d. Perhaps this is complicated by how far downglacier these sites are, but this needs to be elaborated upon. The same is true for the conclusion, where this is discussed. I would suggest removing this from the conclusion.

Specific Comments
*Italics* indicate suggested grammatical changes

L26 - use of "thick" and "thin" is a relative term. I suggest adding in parentheses what constitutes thick and thin.

L35 – consider "*and, when thick, suppresses melt rates.*" or "*and suppresses melt rates when thick.*"

L39 – this sentence is missing its subject, so it's an incomplete sentence. Consider using a semi-colon instead or adding the subject "Alternatively, this anomaly could be caused by…". Also, "or" and "alternatively" are repetitive.

L41 – referring to the debris-cover anomaly here almost across as a result, i.e., Kennicott Glacier experiences the debris cover anomaly. If this is already known, then the reference should be added. If this is not known, then consider changing this sentence to give a broader overview of what's being done, e.g., constrain patterns of … to understand the role of surface melt and ice dynamics on the surface lowering of Kennicott Glacier.

L55 – it's not entirely clear why thinner debris would affect the anomalous glacier thinning explained by melt hotspots, since thinner debris will have melt rates that are closer to clean ice. Also, are there previous debris thickness measurements of Kennicott Glacier? If so, this should be cited; otherwise, the fact that Kennicott Glacier has thinner debris than those previously studied is a result.

L60 – It remains unclear as to why thin debris increases the likelihood that melt hotspots will compensate for the insulating effects of debris. Conversely, the way the argument is stated sounds like melt hotspots cannot compensate for the insulating effects of debris on glaciers; however, because the debris on Kennicott Glacier is thinner, the sub-debris melt rates are closer to clean ice melt rates and hence the melt hotspots are less important because there's less of a difference to compensate for. The key here seems to be more on the sub-debris melt rates of thin debris than the melt hotspots. Please clarify this.

L61 – typo "similar" should be *"similar"*

L72 – typo in the reported elevation range? Also, is there a reference for this data? RGI inventory perhaps?

L73 – consider "… and our study area, the debris-covered tongue of Kennicott Glacier (24.2 km2), is only…"

L77 – be consistent with reporting elevations. Perhaps "Above 700 m a.s.l.". This should be done throughout the manuscript as well, e.g., L90, L131, L134, caption of Figure 1 "located at 1240 m a.s.l.", etc.

L77-79 – is there a reference for these observations?

L86 – What do you mean by "Kennicott Glacier debris"? The debris properties? If so, state this "Because the debris properties of Kennicott Glacier have not been…"

L88 – consider "internal and surface debris temperatures, and …"

Figure 1 – delete the ")" after panel b in the caption. Change to elevations to m a.s.l.  May Creek meteorological station is not shown on the map.  I suggest adding this – perhaps it is covered by one of the legends.

Figure 2 – caption is unclear.  "Dead" ice portion has daily mean surface velocities greater than 5 cm d-1 only during sliding events?  Is this meant to be less than 5 cm d-1 with the exception of sliding events?  Also, what does "and the observations of Rickman and Rosenkrans, 1997" refer to?  Fix this reference.

L107 – Avoid the use of unnecessary acronyms like *LR* for lapse rate. This only makes the manuscript less readable, especially for readers who may not be as familiar with a specific acronym.

Figure 4 – The 4 panel figure is highly repetitive (e.g., shortwave radiation is shown in all 4 panels, and the MWS air temperature is shown in both panels).  I would recommend using only 2 panels. Air temperature can easily show the 3 sites, and the two lapse rates can easily be shown on the same figure by using different colors or styles.

L128 – This line doesn't make sense "at 109 locations at the same locations we also measured". Is it means to be two sentences?  Otherwise, perhaps "around the locations where we measured …".

Table 2 – is 0.001 cm an actual measurement?  That is incredibly precise and thin for a debris thickness, which is hard to believe.

L135-139 - It would be helpful to provide context to the specific sites (panels) for each of these sentences, e.g., "debris thickness did not exceed 15 cm (Fig. 6c)"

L144 – Given the use of MF (used by Pellicciotti et al. 2005) instead of DDF (used by Hock 2003), I would consider either changing the "MF" to "DDF" or add the example citation of Pellicciotti et al. (2005). Note that in some fields MF or DDF could refer to multiplying multiple variables.  I leave it up to the authors as to whether they want to maintain this original convention or adopt newer uses of it (e.g., degree-day factors shown as $f_{ice}$ (Radić and Hock, 2011)).

L148 – Why the use of off-glacier air temperatures when you have data from on-glacier air temperatures?  It would be interesting to see the off-glacier air temperatures over the same period of time – perhaps this could be added to Figure 4 as this would provide some indication of how much the debris warms the air temperature?

L153-157 – Given the impressive amount of data collected, it is disappointing that the authors do not provide a "best-fit" Østrem curve for comparison with other sites.  While there is considerable variability in surface lowering, especially over thin debris that is dependent on local conditions as the authors state, this is clearly something that would affect all previous curves.  Is there a good reason the authors did not do this?  This could be a highly beneficial product for

modelers. If uncertainty is the issue, the authors could easily add uncertainty bounds to the curves.

L176-177 – What does the "mean" debris surface temperature refer to? Is this the mean temperature over the entire study period (at least one week) or was this used to estimate conductivity on a shorter time period? I assume it is the former, but it may be good to be explicit, e.g., "… we then calculate $K_e$ for each temperature profile *over the entire duration of the temperature measurements."*. This would avoid any misunderstandings because the effective thermal conductivity could vary over time, e.g., if there was a change in debris moisture.

Figure 9 – Why is there a point for a debris thickness of 0 with an effective thermal conductivity of 0 W C$_{-1}$ m$_{-1}$? This seems to be unphysical. I also question the "nonlinear" increase in thermal conductivity as a function of debris thickness. There appears to be a fair amount of scatter such that a linear fit might also produce a reasonable fit? Furthermore, if the (0,0) point is discarded, then the linear fit will likely cross the x-axis around 0.4 – 0.5 W C$_{-1}$ W$_{-1}$, which is near the lower range of that estimated based on physical constants (L181; Nicholson and Benn, 2006). Hence, this would be more physically based. Lastly, why is thermal conductivity plotted on the x-axis? The way this is used in the statement seems to be how thermal conductivity varies due to debris thickness and not the other way around. Hence, the debris thickness is the independent variable (typically plotted on the x-axis) and the thermal conductivity is the dependent variable.

L181 – I question "The apparent non-linear increase". See comment above. It would be good to at least see a linear fit as well.

L182 – typo, "*may be* due to…"

L185 – it would be valuable to make assumptions concerning the specific heat capacity and porosity such that a comparison could be shown for the differences in thermal conductivity based on the method.

L206 – type "*were* made…"

L205-206 – were these debris thicknesses already known from the previous debris thickness and ablation stake measurements or were these new measurements? Furthermore, how many "data points" were collected?

L214-218 - why the switch from backwasting rates to backwasting melt factors? It would be easier to read if it were consistent.

Figure 12 – caption, "based *on* the individual melt factor…"

L227 – shouldn't have to restate acronym, although see previous comment about removing it altogether.

L233 – "related *to* the large areas…"

L245-248 – consider changing these sentences so that two sentences in a row don't start with "But…" as this should make it easier to read and understand.

L255 – Please state the percentage of debris thickness measurements that were derived from the top of ice cliffs to provide the reader with some sense of if this was for 50% of 100% of the measurements. "The majority *(X%)* of our debris thickness measurements…"

L278-279 – This sentence about Part B is confusing. What does estimate if ice cliff melt rates correspond to the location of maximum thinning under thick debris on Kennicott Glacier mean? Is "under thick debris" meant to refer to the debris-covered glacier? A specific part of the glacier? Or literally the areas where the debris is thickest? I assume this is generally referring to the debris-covered glacier, but please clarify to avoid confusion.

L281-285 – Is (1) different than (2)? Or is the poor representation of air temperature due to using the off-glacier meteorological data, which does not account for the variations in air temperature above the debris? Also, having sentences in the middle of these various points is very hard to read. I would suggest making these three separate sentences.

L285 – What does this sentence of the portion of fine material have to do with ice cliffs? This seems very out of place and appears to refer to the section on thermal conductivities.

L297 – missing Oxford comma, which seems to be used throughout the rest of the manuscript

L300 – "transverse debris thickness patterns broadly correspond with surface velocities" is out of place and perhaps meant for paper B or C. This paper showed no data on surface velocities.

L302 – may want to acknowledge the limitations that were described in the discussion, i.e., that most debris thickness measurements were from on top of ice cliffs and so caution should be used when using these for tuning and validating distributed debris thickness estimates as they may underestimate the actual debris thickness.

L305 – reconsider "non-linear" relationship. See comment above. Furthermore, is the larger point that "water" or "porosity" plays an important role in heat transfer? They are certainly related to one another, but most of the discussion seemed to focus on the role of finer debris and porosity. This should be consistent in the conclusion.

L308 – there is no evidence in this paper that the ice cliffs counteract the insulating effects of thick debris. More appropriate would be to summarize how the backwasting melt rates compared to the sub-debris melt rates. If this is a conclusion from Part B, then it belongs in that paper.

References
Pellicciotti, F., Brock, B., Strasser, U., Burlando, P., Funk, M., and Corripio, J. (2005). An enhanced temperature-index glacier melt model including the shortwave radiation balance: development and testing for Haut Glacier d'Arolla, Switzerland, *Journal of Glaciology*, 51(175):573-587.

Radić, V. and Hock, R. (2011). Regionally differentiated contribution of mountain glaciers and ice caps to future sea-level rise, *Nature Geoscience*, 4:91-94.

---

## Referee Comment (RC2) · Evan Miles (Referee) · 29 Oct 2019

**Review of 'Debris cover and the thinning of Kennicott Glacier, Alaska, Part A' by Leif Anderson et al., under consideration for *The Cryosphere***

The manuscript by L Anderson, et al., presents a variety of field measurements on debris-covered Kennicott Glacier, and characterises the debris properties and melt rates under debris or at ice cliffs. These data are an extremely useful contribution to understanding of debris covered glaciers in distinct settings. Very few measurements of debris-covered glaciers are available in Alaska, despite the extensive debris coverage of glaciers in the region. The data presented cover an extensive set of topics, and will be useful in calibrating and applying models developed for other regions to Alaskan sites.

Although there are only minor points of criticism relating to the data presented, the manuscript at present lacks cohesion. The results from this manuscript are key in laying the foundation for Parts B and C of the study by Anderson et al, but I can't shake the feeling that this would better fit as (largely) supplementary material for Part B, or as a submission to the EGU journal Earth Systems Science Data; the content is unusual for The Cryosphere. In the latter case or if the manuscript will remain as an independent paper in The Cryosphere, I would recommend expanding the discussion of the varied data collected; some opportunities for expanded discussion are identified in my comments below.

*Major Points*

As a presentation of diverse field measurements, the manuscript lacks a storyline. I appreciate the effort and value of collecting these measurements, but there is no methodological development, and the results and discussion seem geared towards briefly placing the measurements in the context of observations in High Mountain Asia. The few major outcomes (e.g. aspect dependence of ice cliffs) are not investigated or discussed in much detail, as it is very clear that these measurements are geared towards supporting Part B. Consequently, I feel as though many of the results could be included in Part B without a separate Part A; rather by including these measurements as supplementary material, as they follow more-or-less established methods.

The manuscript organisation is awkward at times. In part this is because measurements and results are presented together, but also because figures are not always associated with the text that pertains to them.  More problematic is the lack of an integrating discussion – the individual measurements are discussed but there is not much of a summary characterisation of Kennicott.  I appreciate that this is difficult to do from such diverse field measurements. Again, this is in part because the paper is unusual for content in The Cryosphere, and this is another reason why I think this work could be integrated into Part B (or as a manuscript in the EGU journal Earth Systems Science Data, rather than a distinct manuscript.

Data availability. In the modern spirit of open data, I would strongly recommend that these measurements be archived in an open repository.

Off-glacier air temperatures are used to correct short-period met measurements to the full period of record, but these stations have been shown in this manuscript to represent entirely different altitudinal temperature differences compared to on-glacier stations. The use of the off-glacier stations needs to be robustly evaluated at the stations, and the on-glacier stations need to be used to determine melt factors (for the on-glacier air temperature subperiod). Even if this does not

change the pattern of relative melt factors, this represents a (possibly major) uncertainty in all of the analysis.

Uncertainty in measurements or calculations is not considered at all in the manuscript. Since these measurements are used in two linked following studies, and to draw important conclusions about the dynamics of debris-covered glaciers, I think it is important to frame the results in terms of uncertainty from the start.

*Minor Points*

L34. 'when thick it supresses melt rates' – although common knowledge, it is worthwhile to specify a reference here

L41. Not just explain but also examine; we have evidence of the 'debris-cover anomaly' in High Mountain Asia but not before in Alaska, to my knowledge.

L53. Missing 'glacier' – debris-covered *glacier* mass balance

L55-64. I agree that Kennicott is an interesting case, and a great opportunity to examine the debris-cover anomaly. However, I don't entirely agree with these two justifications in their present form, possibly because a bit more explanation is needed. The presence of thinner debris means that there is less melt enhancement due to cliffs and ponds (ie they may not melt much 'more' than the subdebris ablation), even if their areal coverage is extensive. Your implied point is that the thin debris should lead to less of a melt difference between clean and debris-covered areas, and so the chance of cliffs/ponds/other mechanisms to make up for this is greater. That needs to be made explicit; at present the second rationale is unclear. For the third rationale, it would be beneficial to identify the actual density of ice cliffs in the study area (although this is an output from part B). Readers should not have to jump between the manuscripts to understand the rationale.

L80. The reference to Mount Blackburn does not fit into the text very well – what is the relevance to Kennicott? Debris supply mechanisms? Lithology?

L83. The multiple clauses with commas are a bit awkward.

L88. For consistency, this should be *debris internal* temperature and debris surface temperature.

L93. I suggest changing 'vary' to 'differ'. Boundary layer conditions also vary widely for debris-free glaciers, and for debris-covered glaciers; without a doubt there is overlap in this variability, but the distributions of conditions differ, which is your point.

L106. It would be good to include a very brief description of this important transition, or to simply state that this location is at the base of a prominent bulge. It would also be useful to refer to readers to a more specific area of Part C.

L107. These lapse rates are extremely steep, which makes me wonder if the positions themselves are sufficiently representative of the glacier surface. As elevation tends to be a less direct control on air temperatures over debris, I would recommend fitting the regression to all three observations at once (rather than a 2-step regression). It is highly likely that topographic prominence and proximity to water are both controls on both wind and air temperature over debris (e.g. Shaw and Steiner publications, also Miles et al, 2017 [Frontiers], Supplementary Material).

L114. 'was' should be 'were' as LRs is plural.

L128. It is not clear from Figure 2 which are the 109 locations with debris thickness measurements, as there are more than 109 points when combining sub-debris melt, ice cliff backwasting, and debris temperature.

L130. It would be good to identify these thinner debris positions (especially those with multiple measurements) spatially in Figure 2, rather than just with elevation.

L136. The presentation of these data seems to occur with Figure 7, which is not mentioned here but is quite a jump through the paper.

L140-142. Were repeated subdebris melt measurements made at the same positions? Did the debris thickness change when re-exhuming the stakes? What uncertainty is there in your debris thicknesses or melt rates due to the removal and reburial of debris? (Especially if this occurs repeatedly). A key consideration is that supraglacial debris often presents as sorted, but it is extremely difficult to replace debris in the same state which it was found. This of course is not a problem unique to your measurements, but it should be acknowledged and considered.

L145. This melt factor determination negates SW and LW inputs (and their variability), which may be very important for debris covered glacier surfaces (e.g. Reid, Steiner, Buri ice cliff studies, also Carenzo et al 2016). Although this may not affect your overall results in terms of total melt, it will definitely affect the aspect dependence of subdebris and ice cliff melt. Also, this is clearly determining the mean melt factor for each location; how variable were different melt subperiods for each site?

L148-150. Please explain this estimation of T* more clearly. Are you using the LR between the two off-glacier stations to estimate T* at each location? If so, this estimation needs to be further evaluated relative to the multi-step on-glacier LRs (for the shorter period of measurements for those stations), which differ considerably for the environmental lapse rate. At present, the dependence on off-glacier measurements is not very robust, as your on-glacier air temperature measurements indicate a significant deviation from off-glacier air temperature spatial variability. This will have the effect of smoothing your ice cliff MFs with elevation.

L156-7. This is an interesting comparison, and should be explored a bit in the Discussion. Is this due to latitudinal controls on Ta or SWin? Presumably these glaciers have differing lithologies, and they certainly differ in climatic setting, so perhaps this is a coincidence? I note that there is still a factor of 2 difference between the other glaciers.

L161. This is not shown in Fig 2.

L176. Please justify the use of a linear extrapolation to surface temperature, which differs from interpretation of many debris internal temperature profiles I've seen (often an exponential form is noted when there are sufficient thermistors). It would also be good to include 1-2 plots of the internal temperatures – diurnal variations and means.

L181. I have some qualms with the 'non-linear' increase, which is only because you have imposed (0,0) as an additional point for your fit. Surely, an infinitesimally small debris thickness (which is of course unrealistic) should converge on the thermal conductivity of the rock material itself (i.e. no longer an effective conductivity, but the true conductivity of the material). If you neglect the (0,0) point, this looks most like a linear trend crossing the x-axis at about 0.4 W (C m)$^{-1}$. Also, I think that the non-linearity, if true, needs more consideration and discussion – what are the effects of sorting, for example? Does this imply a bulk density difference between the upper and lower debris layers?

Also, what do you expect conductivity to look like for layers thicker than 1 m (e.g. these would exceed the range estimated by Nicholson and Benn (2006).

L199. It would be good to show the distinct lithological mixes in Figure 9.

L205. Please indicate the accuracy of the Fluke Infrared Thermometer.

L204-208. This section does not clearly follow the past sections, and also does not integrate very well with the rest of the study at present.

L216. Did you classify cliffs based on the presence of streams as well? Part of the results of Brun et al (2016) and others is that any moving water can have the same effect as ponds. In my opinion (not demonstrated) supraglacial streams are even more effective cliff maintenance mechanisms.

L223. It is worth considering these climatological and latitudinal controls in slightly more detail. Is Kennicott really cloudier in the melt season than Lirung (site of Buri and Pellicciott, 2018)? The latitudinal control is not unexpected, but deserves more consideration. Effectively, during the ablation season there should be less diurnal variation in solar zenith angle at high latitude (solar zenith and azimuth are of course correlated seasonally at any latitude).

L233-234. Both instances of 'effected' should be 'affected'.

L264. Are these the (unmodified) measured melt rates or your estimated melt rates from section 2.3?

L265. The comma here is awkward. Perhaps use 'as compared to'

L273. This was only demonstrated for north-facing cliffs in Buri et al (2016b).

L282. I agree that the representation of air temperatures from off-glacier stations is not robust. This deserves careful comparison of estimated air temperatures from lapse rates derived from your on-glacier stations (for the shorter period) before an extrapolation across the glacier. More importantly, this could lead to a major uncertainty in your MFs for both debris and cliffs, even if the patterns do not change with more realistic air temperatures. At the very least an evaluation of the accuracy of the off-glacier stations for representing the on-glacier observed air temperatures is needed.

L304-307. This list of summary statements is not terribly satisfying, and feels like a list of bullet points. More interesting is whether Kennicott's debris properties generally fit within the range of previous distributions (they seem to) which is meaningful as there are few published debris properties in Alaska generally. At the very least, it would be nice to have some numbers in the text?

Table 1. The estimated debris surface temperature difference is not described in the text.

Table 2. I would describe the contents of this table as 'measurements' rather than 'variables'.

Table 3. It seems odd to choose Buri and Pellicciotti (2018) to represent Lirung, as that study was primarily modelling synthetic cliffs rather than reporting backwasting measurements. I think the most appropriate study here would be Brun et al (2016).

Figure 1. At what interval are these contours?

Figure 2. It would be useful to identify the sources and dates of the WV and aerial imagery in this caption or in the text.

Figure 3. I like this schematic, but it's not quite complete: missing are the thermistor strings and air temperature measurements (possibly others). Also, it would be fantastic to include some field photographs demonstrating the measurements.

Figure 4. Since you rely on the May Ck and Gates air temperature measurements, it would be very beneficial to show them here. Perhaps it would also be possible to combine panels (a) and (c), and (b) and (d).

Figure 5. Can you indicate the lithology of the debris thickness in panel (a)?

Figure 6. This seems to be referred to out of place in the text. Also, I'd suggest switching the axes (so that elevation is the y axis) for easier comparison with Figures 1 and 5.

Figure 7. I didn't catch a description of the bare-ice melt rate – what elevation was this at? In addition, this content is almost entirely repeated in Figure 8, so I'd suggest eliminating the figure, but depicting the bare ice melt rate in Figure 8.

Figure 9. As described with my comment on L181, I don't think the point at the origin is justified, in which case a linear fit is entirely appropriate. Also, I'm a bit disappointed that we don't see any of the thermistor data!

Figure 10. I would suggest to merge this with Figure 9, as the content is very closely related. Also, I note that the units here ($m^2 s^{-1}$) differ from that in the text ($mm^2 s^{-1}$).

Figure 11. Over what time period were these temperature measurements taken?

Figure 12. Is it possible to identify the cliffs that bordered ponds or streams within one of these panels?

---

## Author Comment (AC1) · 15 Feb 2020

Reply to Review 1 Part A follows the general response.

Our general response is here:

Thank you, Thank you, Thank you for taking the time to edit and review our manuscripts! We appreciate that this is a large body of work and that many of the reviewers read two manuscripts. And it has been a long process for the editor. We hope we can return the favor to each of you soon.

Below the reviewers and editor will find changes that we would like to make to each of the three parts. These changes are intended to address the issues related largely to the structure of the manuscripts and to make the overall study more rigorous.

**Response to structural critiques**

It is clear that a number of reviewers felt that the manuscripts should be combined in some fashion. While we can understand their perspective we also note that there is more than enough material for 3 manuscripts here. From Reviewer 3 from Part C:

"Moreover I believe that the main messages and findings of the three papers come in separate papers probably better across than in one huge one."

We would prefer that we are given the chance to improve the manuscripts such that a 3 part format works (especially considering the amount of new data, analyses, and conclusions we present across each of the three parts):

- *Publishing one paper after the next?* One possible solution would be to improve each manuscript make each one 'stand alone.' and then publish each manuscript one after the next. Such that Part B could cite Part A, etc. This could avoid the issues about simultaneous publication.
- *Combining the parts into one whole would create a large, complex, less readable paper.* If we were to combine the Parts into one whole the paper would easily have 30 figures, with 3 distinct sets of conclusions. This is the case especially because so little work has been published on debris cover from Alaska.
- *Combining individual parts will change the conclusions we have space to draw from the full body of work.* For this reason we feel that it is best that we are given the chance to make the 3 parts work together. The combination of the papers essentially diminishes, in our mind the impact of each component of the whole. Part A allows us to highlight important melt data that places Kennicott Glacier in context with other debris-covered glaciers, mostly in High Mountain Asia. Part B, introduces a novel ice cliff detection method that performs very well compared to other methods. And shows that Kennicott Glacier has a much higher density of ice cliffs than any other studied debris-covered glacier. Despite having so many ice cliffs they cannot compensate for the melt-reducing effects of debris cover. We feel that the ice cliff detection method is especially important to highlight. The proposal to have one paper would severely limit the space we have to highlight that contribution. Part C highlights new feedbacks that have never been identified before.

- *Combining papers will require that we hide some of our work in the supplemental or that we do not make that data available to other researchers.* Combining the papers requires that we 'hide' some of the work in the supplemental. For example, if we combine Parts A and B the discussion will become too complex to really put the in situ measurements in context with data from other glaciers from other regions.
- *Repetition and needing to look back at other Parts.* Some repetition is unavoidable, but some of the reviewers present a no win scenario. Where there is either too much repetition or not enough. Please see the proposed paragraphs we would add to each part as a means to clarify the structure.
- *Establishing an individual story for Part A*. We feel that building a better story for Part A would solve much of the reviewers' issues. We propose that the main conherent story be placing our mass balance data in context with data from other regions. This would be a vital contribution that places the Kennicott Glacier data in context and establishes a baseline for modelers wanting to go global with their analyses.

We find that the lack of agreement between the reviewers about how to combine the work or leave it the same is dependent on their individual background. For example:

- If the reviewer values in situ data then Part A should stand alone
- If the reviewer values assessing the in situ data while considering the melt extrapolation then Parts A and B should be combined.
- If the reviewer values new methodology then maybe Part C should be combined with Part B.
- If the reviewer values a comparison of melt and ice discharge reduction as the key then perhaps they propose to combine all 3 parts.

Because of the diversity of scholars studying debris-covered glaciers we felt that individual parts would allow different readers to engage the parts of the study that they are most drawn to. In order to make the 3 parts flow together better we propose to add these paragraphs to each manuscript:

"We present three papers which build on one another to address the control of debris on the thinning of the of Kennicott Glacier, Alaska. Each of the three parts builds upon the next. In Part A we present and analyze situ mass balance measurements focusing on the effects of debris and ice cliffs. Because scant mass balance data is available from debris-covered glaciers in Alaska we discuss these measurements in detail and place them in context with measurements from debris-covered glaciers in other regions.

In Part B, we develop a method for delineating ice cliffs and describe how we make distributed mass balance measurements across the debris-covered tongue. Because mass balance estimates, that include the effects of both debris and ice cliffs are rare even where debris-covered glaciers are more commonly studied, we discuss the importance of these estimates in the context of the thinning pattern of Kennicott Glacier.

In Part C, we introduce ice dynamical data which allows us to consider the roles of mass balance, ice dynamics, and surface processes on the thinning of Kennicott glacier. Our analysis uses the continuity equation as a guide. The intention is to treat the debris, ice cliffs, streams, ponds, and ice dynamics as components of a whole that interact to give rise to the thinning pattern of Kennicott Glacier."

The inclusion of these paragraphs will allow readers to move more fluidly between each Part.

**Science critiques**

For a some of the more major science critiques we address them in brief here and in more depth in the individual responses. For other scientific criticisms we refer you to our line-by-line responses to reviewers.

**Melt factor issues related to off-glacier air temperature station (Part A)**

Classical degree day factors are relative to the temperature data used to obtain the melt factors. So there is no absolutely correct meltfactor it is always relative to the temperature data. For energy balance models on-glacier temperature is required but off-glacier temperature is often more representative of the conditions of the atmosphere (they aren't so controlled by the boundary layer effects of the ice).

The melt factor correction is also very small we are simply adjusting data collected over a few weeks to a melt rate over the whole study period of a month and a half or so. If we did not include a melt factor correction our results and conclusions would be virtually the same. We will provide plots to support this in the revision stage.

**Uncertainty analysis (all Parts)**

A generous uncertainty estimate is already in the body of work. In Part B it must not have been clear that the red bands in Figure 10 are extreme bounds on the melt estimates. This needs to be clear. A small percentage of possible outcomes are outside this window. This plot also now includes in increased area when taking into account the slope of ice cliffs (following Evan's review).

We find that this is actually a more elegant way of representing the uncertainty that citing every value in the text with an uncertainty. We will include more uncertanties in Part A though.

*Time coverage dh/dt from 1957 to 2009 (Part B)*

We will present additional dh/dt data that covers the time period of in situ measurement. *Ice discharge change in time*

We will incorporate additional annual surface velocities that show the change in velocity over time so we can estimate the change in ice emergence rate in time.

*Including estimates of melt from ponds, streams, englacial conduits, and the sub-glacial environment (Part B).*

We will include back-of-the-envelope estimates with extreme cases to explore their effect on the thinning pattern.

Part C reveals new feedbacks.

It seemed that some reviewers did not capture the new feedback we are highlighting from Kennicott Glacier. We wonder if their reviews would change if that feedback was better communicated by us?

**Part A: proposed changes**

We feel that there is more than enough new material here for a stand alone paper, but in order to improve the manuscript and create more of a storyline we propose that we add these additional datasets/ideas to Part A:

- Provide error bars for the data, if the plots are too messy we will but the figures with the error bars in the supplemental. Noting that these uncertainties are all less than the extreme uncertainty presented in Figure 10a of Part B.
- A detailed analysis to explain the scatter in the ice cliff backwasting rates and meltfactors.
   Do they correlate with local debris thickness, streams, or lakes?
- Make a comparison of our in situ data with data from else where, likely showing that they are consistent
  - This is important for the important global studies that will be coming out related to debris cover.
  - We will make a broad characterization of the Kennicott Glacier in relation to other glaciers
- Global debris cover anomaly. Highlight that the debris cover anomaly is likely global. We will do this with long profiles of dh/dt from multiple glaciers in the Wrangell Mountains and their debris cover extent. One figure will be added that shows multiple thinning profiles. One table will be added that further shows this. Since the dh/dt data has already been published by Das et al., 2015 we will use this figure as motivation for the individual parts.
- We also have additional data related to the geometry of the ice cliffs that we measured. We will put these data in the supplemental of Part A.
- Add in the paragraph description that links each of the three papers and helps guide the reader through each manuscript.

**Part B: proposed changes**

We feel that there is more than enough new material here for a stand alone paper, but in order to improve the manuscript we propose that we add these additional datasets/ideas to Part B:

- We will add additional text supporting the usefulness of our new ice cliff detection method. In the supplemental we will include additional satellite photos showing how ice cliffs tend to be darker than the surrounding debris so this method can therefore be applied on other glaciers. We will also compare our method with other approaches from other glaciers.
- We will present new DEM differences from 2007 to 2013. These dh/dt data show that the zone of maximum thinning remains in the same spot as for the period from 1957 to 2007. We will also include additional laser altymetry data from 2007 that shows a similar thinning pattern.

-This will address one of the main criticisms from multiple reviewers.

- We will introduce back-of-the-envelope calculations of the possible effect of englacial melt, sub-glacial melt, melt under pond surfaces, and melt by streams. This will clear up any issues related to this manuscript not being comprehensive with regards to melt hotspots.
  - We will not include stream digitizations in this manuscript because we cannot possibly digitize all streams on the glacier surface (imagery is too coarse). The streams play more into the feedbacks in Part C. We will instead make arguments about the surface area coverage of streams and their plausible effect on surface melt.
- We will use a uniform curve fit through the ice cliff backwasting data. And also explore the effect of other curve fits, producing different figure 10a and 10bs which we will put in the supplemental and discuss in the main text.
- Add in the paragraph description that links each of the three papers and helps guide the reader through each manuscript.
- Make sure it is clear how generous the uncertainty estimates already are in this paper. One of the reviewers missed these error estimates completely.
- Emphasize the increasing importance of ice cliffs under thicker and thicker debris.

**Part C: proposed changes**

We want to emphasize here that we do outline new feedbacks in this paper.

From Reviewer 3 from Part C:

"P 2 line 62-63: importantly in part C you not just present data on ice dynamics and supraglacial streams but crucially in part C these data and all components of the mass conservation equation (thinning, flux divergence. . .) are analysed for relation and feedbacks between them. Also say this here, as it is the backbone and most exiting part of this part C."

On Kennicott Glacier there is a strong correspondence between ice cliffs and active ice flow. While weak relationships have been suggested here on Kennicott the correlation is more clear than anywhere else.

The highest concentration of ice cliffs occurs at the upper end of the zone of maximum thinning. The high concentration of ice cliffs also corresponds to where we expect ice emergence rates to be high. These ice emergence rates uplift the glacier surface, working to counter glacier thinning. But ice dynamics, which produce this surface uplift also seems to produce more ice cliffs (see the physical descriptions within the main article). These ice cliffs counter the effect of surface uplift, they are essentially a negative feedback on the effect of ice dynamics.

In addition to this new feedback we also present a number of new hypotheses for the interaction of surface processes with melt and ice dynamics with a new, holistic perspective.

We feel that there is more than enough new material here for a stand alone paper, but in order to improve the manuscript we propose that we add these additional datasets/ideas to Part C:

- New annual surface velocities from 2000-2010

   These velocities allow us to calculate changes in ice emergence rate and ice flux over the in situ measurement period
   More detailed discussion of the reduction of ice emergence rate through time.
- Delineation of drainage basins on the glacier surface (new figure) to support the stream story already within the manuscript.
- Tie in a discussion about glacier surface topography. Ice cliff maximum heights (from in situ measurements), the number of individual ice cliffs with elevation band, and calculated glacier surface relief down glacier.
- New processes drawings to show the important new observations that we are highlighting in this paper. This will greatly improve the reader's ability to see the new process links we are describing.
- Additional photo evidence from the field outlining these new processes links. Many will go into the supplemental but they will support and clarify the process links we are highlighting.
- Description of a new ice cliff burial mechanism. Timelapse movies from the Kennicott and Ngozumpa glaciers (in the supplemental) showing a new mechanism for the burial of ice cliffs. The actual process is not yet described in detail in the text.
- A paragraph that is the same for each of the 3 parts that outlines how they build off of one another.

**Reply to Reviewer 1 Part A**

Review of "Debris cover and the thinning of Kennicott Glacier, Alaska, Part A: in situ mass balance measurements" by Anderson et al.

**Thank you for taking the time to review our manuscripts.**

This study is the first part of three publications that investigate debris cover on Kennicott Glacier in Alaska. Given the limited number of studies that measure properties and melt rates of debris-covered glaciers, these measurements and results are important for advancing our understanding of debris-covered glaciers. This is especially true when one considers the limited knowledge of debris-covered glaciers in Alaska. The measurements and results are presented well. For the most part, the study is easy to follow, well-written, and has sufficient references.

There are a few sentences/paragraphs that could be modified to improve their readability though. The only major comment is to make sure that this study is discussing results that specifically pertain to this part of the three-part study. There are also a couple places where additional detail or analysis would provide useful context to the modeling community; however, this would only require minimal additional work. Therefore, I recommend accepting this manuscript for publication subject to minor revisions. Please see my detailed comments below.

Thank you kindly. We very much appreciate your efforts, especially considering you reviewed two manuscripts.

**Main Comments**

The reasons for studying Kennicott Glacier largely come across as reporting results across the three papers as opposed to stating what each paper does. For example, L55-57 state that the debris is thinner than most previously studied, but there is no reference to any studies concerning debris thicknesses on Kennicott Glacier. Similarly, L58 states there are more ice cliffs than those previously studied without a reference to a study that shows this. Hence, these appear to be results (and results from other papers) that are stated in the introduction.

We did this because the papers are complementary. We feel that in the case of these three papers they come across more coherently if we include some of the justifications from the later papers. We aren't sure this is so bad, especially if we cite those works especially since those results are well supported and reviewed simultaneously. Do these 3 papers need to be completely independent in the way this reviewer suggests? And grow from Part A to B to C? Or can we bring in justification from Parts B and C to A? As we presented here we feel that tying in the justifications from later Parts makes this work more compelling. But we can adjust this if needed.

Furthermore, the introduction states multiple times that the thinner debris increases the likelihood that melt hotspots will compensate for the insulating affects; however, thinner debris has higher melt rates, so it's unclear why melt hotspots would be more important for debris-covered glaciers with thinner debris because there would be less contrast between the sub-debris and ice cliff melt rates. If this is a hypothesis, then please state it this way. If this is supported by a physical basis, then please explicitly state this reasoning.

It is not about the relative contribution of hotspots to sub-debris melt but rather a comparison of absolute melt rates. That is a big point here that we will emphasize better. The absolute melt rate is what matters for the debris covere anomaly.

It is the net melt (sub-debris + hotspots) compared to the bare-ice melt rates at the top of the debris cover. Or another way to put it is: where is the maximum glacier-wide melt rate? And does it correspond with the zone of maximum thinning (concept introduced later in the paper series)

Lastly, the interpretation of the transverse variations of debris thickness appear to be poorly supported by the present figures and text. L135-139 state that mean debris thicknesses increase near the glacier margins. However, site a appears to be closest to the center of the glacier, yet it has thicker debris. Similarly, site c is between sites b and d. Perhaps this is complicated by how far downglacier these sites are, but this needs to be elaborated upon. The same is true for the conclusion, where this is discussed. I would suggest removing this from the conclusion.

We see the reviewer's point. We can also slightly change the wording to highlight that the debris thicknesses are also controlled by the englacial debris concentration and the full strain history of the debris. But that there is a tendency for thickening down glacier and towards the lateral margins.

Specific Comments Italics indicate suggested grammatical changes

L26 - use of "thick" and "thin" is a relative term. I suggest adding in parentheses what constitutes thick and thin.

**We will add this distinction.**

L35 – consider "and, when thick, suppresses melt rates." or "and suppresses melt rates when thick."L39 – this sentence is missing its subject, so it's an incomplete sentence. Consider using a semi-colon instead or adding the subject "Alternatively, this anomaly could be caused by...". Also, "or" and "alternatively" are repetitive.

L41 – referring to the debris-cover anomaly here almost across as a result, i.e., Kennicott Glacier experiences the debris cover anomaly. If this is already known, then the reference should be added. If this is not known, then consider changing this sentence to give a broader overview of what's being done, e.g., constrain patterns of ... to understand the role of surface melt and ice dynamics on the surface lowering of Kennicott Glacier.

We are considering adding an assessment of the debris-cover anomaly in the Wrangell Mountains as part of the results. This would add quite a bit to the study and emphasize the potential global nature of the phenomenon

L55 – it's not entirely clear why thinner debris would affect the anomalous glacier thinning explained by melt hotspots, since thinner debris will have melt rates that are closer to clean ice. Also, are there previous debris thickness measurements of Kennicott Glacier? If so, this should be cited; otherwise, the fact that Kennicott Glacier has thinner debris than those previously studied is a result.

Thank you for the comment, we need to clarify this. If rapid thinning under debris cover is primarily caused by melt (hot spots + sub-debris melt) then we are most likely to see this effect where debris is thin and sub-debris melt rates are high. The basic logic we use throughout the 3 parts is: if melt rates are the primary control on thinning then melt must also be maximized where thinning is greatest. Having thin debris already creates high melt rates, adding a high coverage of ice cliffs means that both components (hot spots and sub-debris melt) are extreme for Kennicott Glacier. We will make this clear in the updated manuscript.

We will move the results to the results section as recommended.

L60 – It remains unclear as to why thin debris increases the likelihood that melt hotspots will compensate for the insulating effects of debris.

It is very simply that thin debris reduces melt less than thick debris relative to hypothetical, local bare ice melt rates. If you have a glacier with thick debris cover melt suppression will be higher relative to hypothetical local bare-ice melt rates and will require a much higher contribution of melt from hot spots to compensate for the insulating effects of debris.

Conversely, the way the argument is stated sounds like melt hotspots cannot compensate for the insulating effects of debris on glaciers; however, because the debris on Kennicott Glacier is thinner, the sub-debris melt rates are closer to clean ice melt rates and hence the melt hotspots are less important because there's less of a difference to compensate for. The key here seems to be more on the sub-debris melt rates of thin debris than the melt hotspots. Please clarify this.

We are happy to clarify this in the text. Thin debris represents an extreme case where melt rates are already higher, adding on a high concentration of ice cliffs means that we are likely to get high melt rates in the debris-covered area.

L61 – typo "similar" should be "similar"

L72 – typo in the reported elevation range? Also, is there a reference for this data? RGI inventory perhaps?

L73 – consider "... and our study area, the debris-covered tongue of Kennicott Glacier (24.2 km 2 ), is only..."

L77 – be consistent with reporting elevations. Perhaps "Above 700 m a.s.l.". This should be done throughout the manuscript as well, e.g., L90, L131, L134, caption of Figure 1 "located at 1240 m a.s.l.", etc.

L77-79 – is there a reference for these observations?

We could cite LSA's dissertation here. Or move this out of the introduction following the other comments.

L86 – What do you mean by "Kennicott Glacier debris"? The debris properties? If so, state this "Because the debris properties of Kennicott Glacier have not been..."

**We will clarify this.**

L88 – consider "internal and surface debris temperatures, and …"Figure 1 – delete the ")" after panel b in the caption. Change to elevations to m a.s.l. May Creek meteorological station is not shown on the map. I suggest adding this – perhaps it is covered by one of the legends.

**We will fix this.**

Figure 2 – caption is unclear. "Dead" ice portion has daily mean surface velocities greater than 5 cm d -1 only during sliding events? Is this meant to be less than 5 cm d -1 with the exception of sliding events? Also, what does "and the observations of Rickman and Rosenkrans, 1997" refer

**to? Fix this reference.**

We will clear this up. But we are following the interpretation of "dead ice" from Rickman and Rosenkrans at this point, which also match slow velocities even when sliding is at it's maximum in the summer following Armstrong et al., 2016.

L107 – Avoid the use of unnecessary acronyms like LR for lapse rate. This only makes the manuscript less readable, especially for readers who may not be as familiar with a specific acronym.

**We will fix this.**

Figure 4 – The 4 panel figure is highly repetitive (e.g., shortwave radiation is shown in all 4 panels, and the MWS air temperature is shown in both panels). I would recommend using only 2 panels. Air temperature can easily show the 3 sites, and the two lapse rates can easily be shown on the same figure by using different colors or styles.

The problem is that the figures become too difficult to read following this suggestion. We are not sure that this is a big issue.

L128 – This line doesn't make sense "at 109 locations at the same locations we also measured". Is it means to be two sentences? Otherwise, perhaps "around the locations where we measured …".

**We will clarify this.**

Table 2 – is 0.001 cm an actual measurement? That is incredibly precise and thin for a debris thickness, which is hard to believe.

**We will clarify this. You are right.**

L135-139 - It would be helpful to provide context to the specific sites (panels) for each of these sentences, e.g., "debris thickness did not exceed 15 cm (Fig. 6c)"

**Helpful thank you.**

L144 – Given the use of MF (used by Pellicciotti et al. 2005) instead of DDF (used by Hock 2003), I would consider either changing the "MF" to "DDF" or add the example citation of Pellicciotti et al. (2005). Note that in some fields MF or DDF could refer to multiplying multiple variables. I leave it up to the authors as to whether they want to maintain this original convention or adopt newer uses of it (e.g., degree-day factors shown as f ice (Radić and Hock, 2011)).

**Also very helpful. We will clarify this and add relevant citations.**

L148 – Why the use of off-glacier air temperatures when you have data from on-glacier air temperatures? It would be interesting to see the off-glacier air temperatures over the same period of time – perhaps this could be added to Figure 4 as this would provide some indication of how much the debris warms the air temperature?

We use the off-glacier air stations because we do not have measurements for the full time period of the field campaign from on glacier. There is a local station in McCarthy but it is not automatic and

is recorded only during work hours for the airport. For melt factors it is also common to use off glacier sites and they actually perform better than on glacier sites often times. The idea is that on glacier sites are affected by the ice surface itself but really what is controlling available energy for melt is the integrated temperature from the lower 1 km of the atmosphere.

In addition to this the meltfactor correction provides a minor correction to the melt rates. We include this to be complete and correct measurements for difference measurement intervals.

L153-157 – Given the impressive amount of data collected, it is disappointing that the authors do not provide a "best-fit" Østrem curve for comparison with other sites. While there is considerable variability in surface lowering, especially over thin debris that is dependent on local conditions as the authors state, this is clearly something that would affect all previous curves. Is there a good reason the authors did not do this? This could be a highly beneficial product for modelers. If uncertainty is the issue, the authors could easily add uncertainty bounds to the curves.

**We can certainly include a curve fit. Along these lines we will also provide a 'debris-covered tongue wide estimate of melt if the area was totally debris covered.**

L176-177 – What does the "mean" debris surface temperature refer to? Is this the mean temperature over the entire study period (at least one week) or was this used to estimate conductivity on a shorter time period? I assume it is the former, but it may be good to be explicit, e.g., "... we then calculate K e for each temperature profile over the entire duration of the temperature measurements.". This would avoid any misunderstandings because the effective thermal conductivity could vary over time, e.g., if there was a change in debris moisture.

**Thank you for this comment. We will clarify this in the text. It is the mean temperature over the entire study period (at least one week).**

Figure 9 – Why is there a point for a debris thickness of 0 with an effective thermal conductivity of 0 W C -1 m -1 ? This seems to be unphysical. I also question the "nonlinear" increase in thermal conductivity as a function of debris thickness. There appears to be a fair amount of scatter such that a linear fit might also produce a reasonable fit? Furthermore, if the (0,0) point is discarded, then the linear fit will likely cross the x-axis around 0.4 - 0.5 W C -1 W -1 , which is near the lower range of that estimated based on physical constants (L181; Nicholson and Benn, 2006). Hence, this would be more physically based. Lastly, why is thermal conductivity varies due to debris thickness and not the other way around. Hence, the debris thickness is the independent variable (typically plotted on the x-axis) and the thermal conductivity is the dependent variable.

These are all good points and we will remove the zero point and reconsider the conclusions and non-linear statements, emphasizing that this is a minor change to the scientific conclusions we draw.

L181 – I question "The apparent non-linear increase". See comment above. It would be good to at least see a linear fit as well.

**We will re-evaluate this and change the text as suggested.**

L182 – typo, "may be due to…"

L185 – it would be valuable to make assumptions concerning the specific heat capacity and porosity such that a comparison could be shown for the differences in thermal conductivity based

on the method.

This is a nice suggestion and we can add a discussion of this in a revised version of the manuscript.

L206 – type "were made..."

L205-206 – were these debris thicknesses already known from the previous debris thickness and ablation stake measurements or were these new measurements? Furthermore, how many "data points" were collected?

L214-218 - why the switch from backwasting rates to backwasting melt factors? It would be easier to read if it were consistent.

Figure 12 – caption, "based on the individual melt factor..."

L227 – shouldn't have to restate acronym, although see previous comment about removing it altogether.

L233 – "related to the large areas..."L245-248 – consider changing these sentences so that two sentences in a row don't start with "But..." as this should make it easier to read and understand.

L255 – Please state the percentage of debris thickness measurements that were derived from the top of ice cliffs to provide the reader with some sense of if this was for 50% of 100% of the measurements. "The majority (X%) of our debris thickness measurements..."

L278-279 – This sentence about Part B is confusing. What does estimate if ice cliff melt rates correspond to the location of maximum thinning under thick debris on Kennicott Glacier mean? Is "under thick debris" meant to refer to the debris-covered glacier? A specific part of the glacier? Or literally the areas where the debris is thickest? I assume this is generally referring to the debris-covered glacier, but please clarify to avoid confusion.

**We will clarify these points.**

L281-285 – Is (1) different than (2)? Or is the poor representation of air temperature due to using the off-glacier meteorological data, which does not account for the variations in air temperature above the debris? Also, having sentences in the middle of these various points is very hard to read. I would suggest making these three separate sentences.

We see what the reviewer means. This could be clarified with a bit more explanation.

L285 – What does this sentence of the portion of fine material have to do with ice cliffs? This seems very out of place and appears to refer to the section on thermal conductivities.

This section is actually about ice cliffs (3.4 Ice cliff backwasting). Just needs a bit more of a clear explanation. We will add this text but the point is that finer debris is better able to stay on ice cliff surfaces while more coarse debris trundles down the ice cliff. The more fine material in the debris cover the more-likely that fine material is to adhere to the ice cliff surface. We will add photos of this effect to the paper to emphasize this potentially important control on backwasting rates.

L297 – missing Oxford comma, which seems to be used throughout the rest of the manuscript

L300 – "transverse debris thickness patterns broadly correspond with surface velocities" is out of

place and perhaps meant for paper B or C. This paper showed no data on surface velocities.

We can also make a more hypothetical statement here.

L302 – may want to acknowledge the limitations that were described in the discussion, i.e., that most debris thickness measurements were from on top of ice cliffs and so caution should be used when using these for tuning and validating distributed debris thickness estimates as they may underestimate the actual debris thickness.

**We will consider this suggestion further.**

L305 – reconsider "non-linear" relationship. See comment above. Furthermore, is the larger point that "water" or "porosity" plays an important role in heat transfer? They are certainly related to one another, but most of the discussion seemed to focus on the role of finer debris and porosity. This should be consistent in the conclusion.

We agree and will re analyze the conductivity conclusion.

L308 – there is no evidence in this paper that the ice cliffs counteract the insulating effects of thick debris. More appropriate would be to summarize how the backwasting melt rates compared to the sub-debris melt rates. If this is a conclusion from Part B, then it belongs in that paper.

This just requires a subtle change in the wording or how the sentence is read. But we will not bring in results from Parts B or C to this manuscript.

References (thanks for including these. Very kind)

Pelliciotti, F., Brock, B., Strasser, U., Burlando, P., Funk, M., and Corripio, J. (2005). An enhanced temperature-index glacier melt model including the shortwave radiation balance: development and testing for Haut Glacier d'Arolla, Switzerland, Journal of Glaciology, 51(175):573-587.

Radić, V. and Hock, R. (2011). Regionally differentiated contribution of mountain glaciers and ice caps to future sea-level rise, Nature Geoscience, 4:91-94.

In addition to the changes proposed in the text above we propose these changes:

**Part A: proposed changes**

We feel that there is more than enough new material here for a stand alone paper, but in order to improve the manuscript and create more of a storyline we propose that we add these additional datasets/ideas to Part A:

- Provide error bars for the data, if the plots are too messy we will but the figures with the error bars in the supplemental. Noting that these uncertainties are all less than the extreme uncertainty presented in Figure 10a of Part B.
- A detailed analysis to explain the scatter in the ice cliff backwasting rates and meltfactors.
   Do they correlate with local debris thickness, streams, or lakes?
- Make a comparison of our in situ data with data from else where, likely showing that they are consistent

- This is important for the important global studies that will be coming out related to debris cover.
- We will make a broad characterization of the Kennicott Glacier in relation to other glaciers
- Global debris cover anomaly. Highlight that the debris cover anomaly is likely global. We will do this with long profiles of dh/dt from multiple glaciers in the Wrangell Mountains and their debris cover extent. One figure will be added that shows multiple thinning profiles. One table will be added that further shows this. Since the dh/dt data has already been published by Das et al., 2015 we will use this figure as motivation for the individual parts.
- We also have additional data related to the geometry of the ice cliffs that we measured. We will put these data in the supplemental of Part A.
- Add in the paragraph description that links each of the three papers and helps guide the reader through each manuscript.

---

## Author Comment (AC2) · 15 Feb 2020

Review of 'Debris cover and the thinning of Kennicott Glacier, Alaska, Part A' by Leif Anderson et al., under consideration for The Cryosphere

Thank you kindly for taking the time to review our manuscripts.

The manuscript by L Anderson, et al., presents a variety of field measurements on debris-covered Kennicott Glacier, and characterises the debris properties and melt rates under debris or at ice cliffs. These data are an extremely useful contribution to understanding of debris covered glaciers in distinct settings. Very few measurements of debris-covered glaciers are available in Alaska, despite the extensive debris coverage of glaciers in the region. The data presented cover an extensive set of topics, and will be useful in calibrating and applying models developed for other regions to Alaskan sites.

Although there are only minor points of criticism relating to the data presented, the manuscript at present lacks cohesion. The results from this manuscript are key in laying the foundation for Parts B and C of the study by Anderson et al, but I can't shake the feeling that this would better fit as (largely) supplementary material for Part B, or as a submission to the EGU journal Earth Systems Science Data; the content is unusual for The Cryosphere. In the latter case or if the manuscript will remain as an independent paper in The Cryosphere, I would recommend expanding the discussion of the varied data collected; some opportunities for expanded discussion are identified in my comments below.

Major Points
As a presentation of diverse field measurements, the manuscript lacks a storyline. I appreciate the effort and value of collecting these measurements, but there is no methodological development, and the results and discussion seem geared towards briefly placing the measurements in the context of observations in High Mountain Asia. The few major outcomes (e.g. aspect dependence of ice cliffs) are not investigated or discussed in much detail, as it is very clear that these measurements are geared towards supporting Part B. Consequently, I feel as though many of the results could be included in Part B without a separate Part A; rather by including these measurements as supplementary material, as they follow more-or-less established methods.

The manuscript organisation is awkward at times. In part this is because measurements and results are presented together, but also because figures are not always associated with the text that pertains to them. More problematic is the lack of an integrating discussion – the individual measurements are discussed but there is not much of a summary characterisation of Kennicott. I appreciate that this is difficult to do from such diverse field measurements. Again, this is in part because the paper is unusual for content in The Cryosphere, and this is another reason why I think this work could be integrated into Part B (or as a manuscript in the EGU journal Earth Systems Science Data, rather than a distinct manuscript.

Data availability. In the modern spirit of open data, I would strongly recommend that these measurements be archived in an open repository.

Off-glacier air temperatures are used to correct short-period met measurements to the full period of record, but these stations have been shown in this manuscript to represent entirely different altitudinal temperature differences compared to on-glacier stations. The use of the off-glacier stations needs to be robustly evaluated at the stations, and the on-glacier stations need to be used to determine melt factors (for the on-glacier air temperature subperiod). Even if this does notchange the pattern of relative melt factors, this represents a (possibly major) uncertainty in all of the

analysis.

Uncertainty in measurements or calculations is not considered at all in the manuscript. Since these measurements are used in two linked following studies, and to draw important conclusions about the dynamics of debris-covered glaciers, I think it is important to frame the results in terms of uncertainty from the start.

Minor Points
L34. 'when thick it supresses melt rates' – although common knowledge, it is worthwhile to specify a reference here

We will add a citation here.

L41. Not just explain but also examine; we have evidence of the 'debris-cover anomaly' in High Mountain Asia but not before in Alaska, to my knowledge.

Here, is an opportunity for us to add more to this manuscript and make it more of a stand-alone contribution as this reviewer prefers. We can add in DEM-differencing data after Das, et al., 2015 to show that the debris-cover anomaly is occurring in AK, and is therefore potentially a global effect. If we added this the manuscript would could have a more coherent story line, despite the fact that much of the data lays a foundation for further DC modeling studies (see David's review of Part A) as well as Parts B and C.

L53. Missing 'glacier' – debris-covered glacier mass balance

L55-64. I agree that Kennicott is an interesting case, and a great opportunity to examine the debris-cover anomaly. However, I don't entirely agree with these two justifications in their present form, possibly because a bit more explanation is needed. The presence of thinner debris means that there is less melt enhancement due to cliffs and ponds (ie they may not melt much 'more' than the subdebris ablation), even if their areal coverage is extensive. Your implied point is that the thin debris should lead to less of a melt difference between clean and debris-covered areas, and so the chance of cliffs/ponds/other mechanisms to make up for this is greater. That needs to be made explicit; at present the second rationale is unclear.

Thank you for pointing this out and David's review has a similar comment, and we certainty need to address this. The point is that with thinner debris and more ice cliffs the net melt rate will be closer to the bare ice melt rate not that hot spots will contribute a higher percentage of melt. What really matters for the thinning of the glacier is the absolute melt rate not the relative effect of hotspots as has been emphasized in the literature.

For the third rationale, it would be beneficial to identify the actual density of ice cliffs in the study area (although this is an output from part B).

We would like to add the actual density of ice cliffs in the study area and we will, but David's review commented that we need to not have results from Parts B and C in the introduction of A. But we simply feel that these contributions are components of a whole and that since these works are being reviewed simultaneously that we can provide full justification from the main conclusions from the other parts.

Readers should not have to jump between the manuscripts to understand the rationale.

 We agree and would like to include the results you suggest.

L80. The reference to Mount Blackburn does not fit into the text very well – what is the relevance to Kennicott? Debris supply mechanisms? Lithology?

We will correct this and make its relevance more clear.

L83. The multiple clauses with commas are a bit awkward.

L88. For consistency, this should be debris internal temperature and debris surface temperature.

L93. I suggest changing 'vary' to 'differ'. Boundary layer conditions also vary widely for debris-free glaciers, and for debris-covered glaciers; without a doubt there is overlap in this variability, but the distributions of conditions differ, which is your point.

L106. It would be good to include a very brief description of this important transition, or to simply state that this location is at the base of a prominent bulge. It would also be useful to refer to readers to a more specific area of Part C.

Yes we will include a more detailed description where to look in Part C. We would like to apply this concept through all 3 parts. This will make the three parts mesh better together.

L107. These lapse rates are extremely steep, which makes me wonder if the positions themselves are sufficiently representative of the glacier surface. As elevation tends to be a less direct control on air temperatures over debris, I would recommend fitting the regression to all three observations at once (rather than a 2-step regression).

This is a very helpful suggestion thank you we will implement this.

It is highly likely that topographic prominence and proximity to water are both controls on both wind and air temperature over debris (e.g. Shaw and Steiner publications, also Miles et al, 2017 [Frontiers], Supplementary Material).

These are great suggestions, thank you! Here we really highlight the proximity to wide stretches of bare ice, which is the case on the Kennicott, unlike many previously studied DCG. We will discuss these though in the revised manuscript though. We agree that this should be discussed.

L114. 'was' should be 'were' as LRs is plural.L128. It is not clear from Figure 2 which are the 109 locations with debris thickness measurements, as there are more than 109 points when combining sub-debris melt, ice cliff backwasting, and debris temperature.

We will make this more clear in the revisions. Adding in more clear maps.

L130. It would be good to identify these thinner debris positions (especially those with multiple measurements) spatially in Figure 2, rather than just with elevation.

Yes, we can do this if it won't make the figure too complex. We will try.

L136. The presentation of these data seems to occur with Figure 7, which is not mentioned here but is quite a jump through the paper.

True we will fix this issue.

L140-142. Were repeated subdebris melt measurements made at the same positions? Did the debris thickness change when re-exhuming the stakes? What uncertainty is there in your debris thicknesses or melt rates due to the removal and reburial of debris? (Especially if this occurs repeatedly). A key consideration is that supraglacial debris often presents as sorted, but it is extremely difficult to replace debris in the same state which it was found. This of course is not a problem unique to your measurements, but it should be acknowledged and considered.

Yes this is an issue to all sub-debris melt measurements. We only dug a couple of poles more than twice, but we can emphasize this in the manuscript.

L145. This melt factor determination negates SW and LW inputs (and their variability), which may be very important for debris covered glacier surfaces (e.g. Reid, Steiner, Buri ice cliff studies, also Carenzo et al 2016). Although this may not affect your overall results in terms of total melt, it will definitely affect the aspect dependence of subdebris and ice cliff melt. Also, this is clearly determining the mean melt factor for each location; how variable were different melt subperiods for each site?

To us, the melt factor (MF) approach we already use includes these aspect effects. If the melt is higher in a southerly direction then the MF would be higher. If a north facing ice cliff retreats slower than the MF would be lower. The SW and LW effects may be able to produce more accurate estimates of melt but that would play more of a role if cloudiness changed and the relative effect of SW to LW fluxes changes. But a simple MF approach would also include these effects if the relative effect of SW to LW changes as well.

Our approach here is not to use the most sophisticated melt model possible, that requires may more data input (we are in a relatively data poor region for glacier studies and have no access to these fluxes locally) and increased constraint of parameters. The simple approach used here is effective, and please note how small the corrections are in figure 12. It won't matter which melt model we use the change will be small because we are correcting the rates for a difference of a few weeks between individual measurements. But the differences between melt models is a worthy target of research.

L148-150. Please explain this estimation of T* more clearly. Are you using the LR between the two off-glacier stations to estimate T* at each location? If so, this estimation needs to be further evaluated relative to the multi-step on-glacier LRs (for the shorter period of measurements for those stations), which differ considerably for the environmental lapse rate.

Yes we are using two off glacier meteorological stations which has been shown to provide good estimates of melt. As far as we understand off glacier sites provide a better sense of the temperature of the lowest km of the atmosphere which works well for predicting melt. Note too what we use these MFs for: It is just to correct difference in measurement period so we are deriving MF from a couple of weeks to estimate melt for another couple of weeks. The effect on backwasting rates is very small and does not change the story here or in part B, even if we used a more complex model we aren't convinced that anything would change.

At present, the dependence on off-glacier measurements is not very robust, as your on-glacier air temperature measurements indicate a significant deviation from off-glacier air temperature spatial variability. This will have the effect of smoothing your ice cliff MFs with elevation.

We do not see how this really matters. We use the closest, viable meteorological station. The difference in temperatures observed from the on-glacier stations will be included in differences in the MF between sites. See Flowers article on the use of MF from Canada. That is the advantage of

the MF approach. The MF includes all these differences in physical variability. Even if an Energy Balance model includes all the Energy transfer pathways the number of parameters skyrockets such that the issue becomes the constraint of these parameters.  If we were using an EB approach then yes we need on glacier temperatures, but we are not and we feel that based on a number of studies this approach is justified. We are happy to further discuss this but are not sure that this really effects our study or Parts B or C.

We could do an analysis of MF in space across the glacier though.

L156-7. This is an interesting comparison, and should be explored a bit in the Discussion. Is this due to latitudinal controls on Ta or SWin? Presumably these glaciers have differing lithologies, and they certainly differ in climatic setting, so perhaps this is a coincidence? I note that there is still a factor of 2 difference between the other glaciers.

We agree that this is interesting and will expand on it in the discussion. But it is not clear that we can pull out causality based on available data.

L161. This is not shown in Fig 2.

We will add the locations.

L176. Please justify the use of a linear extrapolation to surface temperature, which differs from interpretation of many debris internal temperature profiles I've seen (often an exponential form is noted when there are sufficient thermistors).

When integrating over more than a week the temperature profile becomes linear when heat is transferred by conduction (Conway and Rassmussen, 2000). We will emphasize this here.

It would also be good to include 1-2 plots of the internal temperatures – diurnal variations and means.

Yes we can include this.

L181. I have some qualms with the 'non-linear' increase, which is only because you have imposed (0,0) as an additional point for your fit. Surely, an infinitesimally small debris thickness (which is of course unrealistic) should converge on the thermal conductivity of the rock material itself (i.e. no longer an effective conductivity, but the true conductivity of the material). If you neglect the (0,0) point, this looks most like a linear trend crossing the x-axis at about 0.4 W (C m) -1 . Also, I think that the non-linearity, if true, needs more consideration and discussion – what are the effects of sorting, for example? Does this imply a bulk density difference between the upper and lower debris layers?Also, what do you expect conductivity to look like for layers thicker than 1 m (e.g. these would exceed the range estimated by Nicholson and Benn (2006).

We will take another look at this and the literature as well. But we agree that we need to be sure here and remove the 0, 0 point. But in the end this does not have major implications for the rest of the work, rather for future melt modelers.

L199. It would be good to show the distinct lithological mixes in Figure 9.

Hmm… we did not do a distinct analysis of lithology but we can extrapolate from geologic maps and our field descriptions.

L205. Please indicate the accuracy of the Fluke Infrared Thermometer.

We will add this.

L204-208. This section does not clearly follow the past sections, and also does not integrate very well with the rest of the study at present.

Thanks for pointing this out. We can add context here to explain why we took these measurements.

L216. Did you classify cliffs based on the presence of streams as well? Part of the results of Brun et al (2016) and others is that any moving water can have the same effect as ponds. In my opinion (not demonstrated) supraglacial streams are even more effective cliff maintenance mechanisms.

We did take notes on the presence of streams at the base and will add that to the manuscript.

L223. It is worth considering these climatological and latitudinal controls in slightly more detail. Is Kennicott really cloudier in the melt season than Lirung (site of Buri and Pellicciott, 2018)? The latitudinal control is not unexpected, but deserves more consideration. Effectively, during the ablation season there should be less diurnal variation in solar zenith angle at high latitude (solar zenith and azimuth are of course correlated seasonally at any latitude).

We will consider it but there are a number of variables to consider here to make conclusive statements. We will try but not dwell on this point.

L233-234. Both instances of 'effected' should be 'affected'.

L264. Are these the (unmodified) measured melt rates or your estimated melt rates from section 2.3?

It doesn't matter which. We will include a plot of measured versus corrected melt rates in the supplemental. The points virtually plot ontop of one another the changes are tiny.

L265. The comma here is awkward. Perhaps use 'as compared to'

L273. This was only demonstrated for north-facing cliffs in Buri et al (2016b).

L282. I agree that the representation of air temperatures from off-glacier stations is not robust. This deserves careful comparison of estimated air temperatures from lapse rates derived from your on-glacier stations (for the shorter period) before an extrapolation across the glacier. More importantly, this could lead to a major uncertainty in your MFs for both debris and cliffs, even if the patterns do not change with more realistic air temperatures. At the very least an evaluation of the accuracy of the off-glacier stations for representing the on-glacier observed air temperatures is needed.

We disagree. There is no need for the off glacier temperature to be compared to on glacier sites. MF are relative parameters only relevant to the air temperature measurements. In addition we could also not do the MF correction and the melt rate results would be almost the same.

As long as the MF derived from the air temperature data we used is then used with air temperatures from the same stations the MF extrapolation is viable. We feel that this point is over emphasized. MF and lapse rates are only relative to the temperatures at station.

See Wheler et al., 2014: Effects of Temperature Forcing Provenance and Extrapolation on the Performance of an Empirical Glacier-Melt Model

L304-307. This list of summary statements is not terribly satisfying, and feels like a list of bullet points. More interesting is whether Kennicott's debris properties generally fit within the range of previous distributions (they seem to) which is meaningful as there are few published debris properties in Alaska generally. At the very least, it would be nice to have some numbers in the text?

We can add more context for the measurements made here with other studies globally. And emphasize that the results are not so different from other regions, if that is indeed the case. We note that these are the first measurements of this kind made in Alaska to date.

Table 1. The estimated debris surface temperature difference is not described in the text.

We will add that in.

Table 2. I would describe the contents of this table as 'measurements' rather than 'variables'.

We will change this.

Table 3. It seems odd to choose Buri and Pellicciotti (2018) to represent Lirung, as that study was primarily modelling synthetic cliffs rather than reporting backwasting measurements. I think the most appropriate study here would be Brun et al (2016).

Ok we will change this.

Figure 1. At what interval are these contours?

Figure 2. It would be useful to identify the sources and dates of the WV and aerial imagery in this caption or in the text.

Figure 3. I like this schematic, but it's not quite complete: missing are the thermistor strings and air temperature measurements (possibly others). Also, it would be fantastic to include some field photographs demonstrating the measurements.

Figure 4. Since you rely on the May Ck and Gates air temperature measurements, it would be very beneficial to show them here. Perhaps it would also be possible to combine panels (a) and (c), and (b) and (d).

We feel that combining the panel will make an unintelligible figure. We could add in the off-glacier data but we aren't sure how it really matters. MF are all relative to the temperature data they are derived from as long as data from the same stations used to derive the MF is used for extrapolation the principle holds. There is no absolute MF.

Figure 5. Can you indicate the lithology of the debris thickness in panel (a)?

We did not do a detailed lithology analysis, though we noted major differences in lithology.

Figure 6. This seems to be referred to out of place in the text. Also, I'd suggest switching the axes (so that elevation is the y axis) for easier comparison with Figures 1 and 5.

We could also just switch the x-axis to distance. But we will take a look.

Figure 7. I didn't catch a description of the bare-ice melt rate – what elevation was this at? In addition, this content is almost entirely repeated in Figure 8, so I'd suggest eliminating the figure, but depicting the bare ice melt rate in Figure 8.

We will clarify bare-ice melt rate. We could just make this a 2 panel figure 7 and 8 but this is a very minor change. If figure 7 was dropped it would very easy to desire a panel that is not comparative.

Figure 9. As described with my comment on L181, I don't think the point at the origin is justified, in which case a linear fit is entirely appropriate. Also, I'm a bit disappointed that we don't see any of the thermistor data!

We will re consider the non-linear increase as suggested. We are happy to show temperature profile data but we aren't sure what it adds. But we can show the exponential funnel!

Figure 10. I would suggest to merge this with Figure 9, as the content is very closely related. Also, I note that the units here (m 2 s -1 ) differ from that in the text (mm 2 s -1 ).

Also ok to keep them separate.

Figure 11. Over what time period were these temperature measurements taken?

From 10 am to 4 pm. We will add that in the caption.

Figure 12. Is it possible to identify the cliffs that bordered ponds or streams within one of these panels?

Yes would be neat to see and include.

In addition to the changes proposed above:

**Part A: proposed changes**

We feel that there is more than enough new material here for a stand alone paper, but in order to improve the manuscript and create more of a storyline we propose that we add these additional datasets/ideas to Part A:

- Provide error bars for the data, if the plots are too messy we will but the figures with the error bars in the supplemental. Noting that these uncertainties are all less than the extreme uncertainty presented in Figure 10a of Part B.
- A detailed analysis to explain the scatter in the ice cliff backwasting rates and meltfactors.
  - Do they correlate with local debris thickness, streams, or lakes?

- Make a comparison of our in situ data with data from else where, likely showing that they are consistent
  - This is important for the important global studies that will be coming out related to debris cover.
  - We will make a broad characterization of the Kennicott Glacier in relation to other glaciers

- Global debris cover anomaly. Highlight that the debris cover anomaly is likely global. We will do this with long profiles of dh/dt from multiple glaciers in the Wrangell Mountains and their debris cover extent. One figure will be added that shows multiple thinning profiles. One table will be added that further shows this. Since the dh/dt data has already been published by Das et al., 2015 we will use this figure as motivation for the individual parts.

- We also have additional data related to the geometry of the ice cliffs that we measured. We will put these data in the supplemental of Part A.

- Add in the paragraph description that links each of the three papers and helps guide the reader through each manuscript.